# LLM-ABBA: Fine-Tuning Large Language Models For Time Series Using Symbolic Approximation

## Abstract

The success of large language models (LLMs) for the time series domain has been demonstrated through various benchmarks. Utilizing symbolic time series representation, one can efficiently bridge the gap between LLMs and time series. However, the remaining challenge is to exploit the semantic information hidden in time series by using symbols or existing tokens of LLMs, while aligning the embedding space of LLMs according to the domain hidden information of time series. The symbolic time series approximation method called ABBA shows outstanding efficacy in preserving salient time series features by modeling time series patterns in terms of amplitude and period while using existing tokens of LLMs.

In this paper, we introduce a method, called LLM-ABBA, that integrates ABBA into large language models for various time series downstream tasks. By symbolizing time series, LLM-ABBA compares favorably to the recent state-of-the-art (SOTA) in UCR and three medical time series classification tasks. Meanwhile, a fixed-polygonal chain trick in ABBA is introduced to avoid large vibrations during prediction tasks by significantly mitigating the effects of cumulative error arisen from misused symbols during the transition from symbols to numerical values. In time series regression tasks, LLM-ABBA achieves the new SOTA on Time Series Extrinsic Regression (TSER) benchmarks. LLM-ABBA also shows competitive prediction capability compared to recent time series prediction SOTA results. We believe this framework can also seamlessly extend to other time series domains.

## 1 Introduction

Time series are fundamental mathematical objects with applications across diverse disciplines such as classification (Ismail Fawaz et al., 2019), regression (Tan et al., 2021), and prediction (Ismail et al., 2020). Recently, the power of large language models (LLMs) in time series applications has been recognized. One review work concludes that there are three main LLM-based approaches to learn intricate semantic and knowledge representations from time series to perform various tasks (Jin et al., 2024). The first approach is to patch and tokenize numerical signals and related text data, followed by fine-tuning on time series tasks (Nie et al., 2022; Jin et al., 2023; Wang et al., 2024);

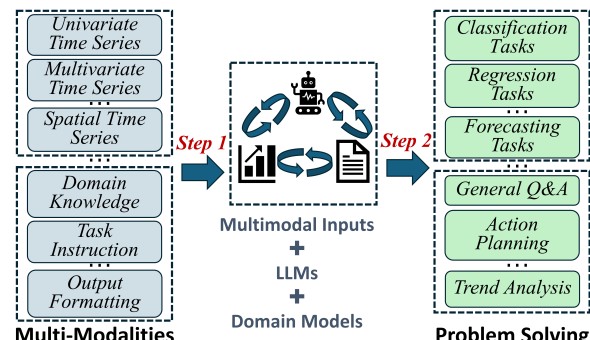

Figure 1: The integration of time series and LLM demonstrates potential in solving complex real-world problems.

the second one is preprocessing time series data to fit LLM input spaces by adding a customized Tokenizer (Gruver et al., 2024); the last one is to build foundation models from scratch, and this approach aims to create large, scalable models, both generic and domain-specific (Rasul et al., 2023; Ekambaram et al., 2024). However, these three techniques each come with their own limitations. Patching and tokenizing time series

segments can build the mapping between time series and the latent embedding of LLMs, instead of discrete language tokens. When outputting the numerical value, this method should generate the digit one by one, which eventually reduces the generation speed (Jin et al., 2023). Furthermore, by adding a customized Tokenizer, LLMs can handle positions of time series patterns and reproduce the internal logic of given time series signals (Mirchandani et al., 2023). Because LLM tokenizers, not designed for numerical values, separate continuous values and ignore the temporal relationship of time series, this method should convert tokens into flexible continuous values (Spathis & Kawsar, 2024). It inevitably requires token transitions from time series feature space to the latent embedding space of LLMs and cannot avoid the risk of semantic loss. Building foundational time series models from scratch can essentially solve these problems. But considering that one should balance the high development costs and their applicability, the challenge of expensive training persists and should be tackled (Jin et al., 2024).

By aligning time series and native language, large language and specialized time series models constitute a new paradigm, where the LLMs are prompted with both time series and text-based instructions (Jin et al., 2024). In this paradigm, time series and textual information provide essential contexts, LLMs contribute to internal knowledge and reasoning capabilities, and time series models offer fundamental pattern recognition assurances. This novel integration is depicted in Figure 1, where a successful combination of these components showcases the potential for a general-purpose, unified system in next-generation time series analysis. Therefore, the challenge is to develop one tool that can transform the internal patterns of time series to the contents that LLMs can recognize (*Step 1* of Figure 1). Moreover, this tool should also transform the generated contents back to the time series domain so as to aid the time series analysis (*Step 2* of Figure 1).

Symbolic time series approximation is a method that converts time series into symbols. It establishes a bridge between strings and numerical time series, which enables the chain-of-pattern (COP) of strings to be as informative as possible compared to raw data. Utilizing the symbolic representation of time series, one can model time series as native languages by encoding time series as a sequence of strings and performing efficient text analysis techniques upon it rather than manipulating raw numerical values, e.g., converting time series forecasting to next-token prediction in text. Symbolic time series approximation could both implicitly and explicitly align the time series features with symbols, which empowers the manipulation of natural language processing learning on time series. If possible, there is no necessity to (1) patch and tokenize time series segments, (2) add an extra customized Tokenizer, or (3) build foundational time series models from scratch. Symbolic representations obtained from transformed numerical time series can potentially reveal the linguistic logic hidden inside time series signals, and this technology roadmap is able to provide LLMs with the ability to understand temporal patterns. Therefore, the time series semantic information can be well exploited in LLMs. Inspired by this idea, it is desirable to obtain a method that can efficiently transform numerical time series into symbols, and fine-tune LLMs on time series analysis tasks (e.g., classification, regression, and prediction).

In this paper, we propose LLM-ABBA, which can help LLMs understand time series by using an adaptive Brownian bridge-based symbolic aggregation (ABBA) method and transforming numerical time series signals into symbolic series. Concretely, LLM-ABBA first transforms time series signals to compressed representations by adaptively compressing numerical inputs. Next, it digitizes the compressed representation with given symbols or pretrained tokens. Then, LLM-ABBA gives LLMs a series of symbols (or pretrained tokens) that LLMs can recognize from the beginning, and these symbols (or pretrained tokens) essentially contain the COP of time series signals. By using the QLoRA fine-tuning method (Dettmers et al., 2024), LLM-ABBA exhibits a trade-off between task performance and efficiency. Finally, to predict the future time series values, LLM-ABBA inversely symbolizes the LLM-generated symbolic representation back to numerical values as predicted values. Therefore, the LLM is capable of incorporating the COP of time series and diving into the analysis of time series on a macroscopic view along with the knowledge from prompting instructive commands.

Our contributions are summarized as follows:

1. We propose a unified and enhanced ABBA approach towards efficiently symbolizing multiple time series and mitigating the accumulated shift in time series reconstruction, enabling an effective inference task over out-of-sample data.

2. LLM-ABBA framework for time series regression tasks achieves a new SOTA, and it also gets a comparable performance on medical time series classification tasks. To the best of our knowledge, this is the first work to combine LLM with the symbolic approximation method for time series.

3. LLM-ABBA can keep the language semantics and learn the COPs of time series by adapter fine-tuning methods in time series forecasting tasks.

4. The universality and convenience of LLMs' multi-modality on time series tasks obtains a valuable improvement.

## 2 RELATED WORK

LLMs for time series methods have made significant achievements in recent years. Gruver et al. (2024) argues that this success stems from the ability of LLMs to naturally represent multimodal distributions of time series, and demonstrates zero-shot generalization abilities of LLMs. By framing a time series forecasting task as a sentence-to-sentence task, AutoTimes (Liu et al., 2024b) minimizes the tunable parameters to generate time series embeddings while freezing the parameters of the LLM, and FPT (Zhou et al., 2023) fine-tunes LLM parameters to serve as a general representation extractor for various time series analysis tasks. These approaches maximize the use of inherent token transitions, leading to improved model efficiency. In terms of multivariate time series forecasting, UniTime (Liu et al., 2024a) trains and fine-tunes a language model to provide a unified forecasting framework across multiple time series domains. Leveraging advanced prompting designs and techniques, PromptCast (Xue & Salim, 2023) transforms time series data into text pairs, and TEMPO (Cao et al., 2023) models specific time series patterns, such as trends and seasonality, by using weighted scatterplot smoothing (Cleveland et al., 1990).

Tuning-based predictors use accessible LLM parameters, typically involving pre-processing and tokenizing numerical signals and related prompt text, followed by fine-tuning on time series tasks (Jin et al., 2024). In summary, there are four steps formulated to adapt LLM to time series:

(i) $\mathcal{T}_{\mathrm{inp}} = \mathrm{Pre\text{-}processing}(\mathcal{T})$: With a Patching operation (Nie et al., 2022; Liu et al., 2024b) or a weighted scatterplot smoothing processing (Cao et al., 2023), time series set $\mathcal{T}$ is pre-processed to specific knowledge-contained inputs $\mathcal{T}_{\mathrm{inp}}$;

(ii) $\mathcal{M}_{\mathrm{inp}} = \mathrm{Tokenizer}(\mathrm{Prompt}, \mathcal{T}_{\mathrm{inp}})$: An additional option is to perform a Tokenizer operation on time series $\mathcal{T}_{\mathrm{inp}}$ and related prompt text to form text sequence tokens $\mathcal{M}_{\mathrm{inp}}$;

(iii) $\mathcal{M}_{\mathrm{outp}} = f_{\mathrm{LLM}}^{\Delta}(\mathcal{M}_{\mathrm{inp}})$: With the instruction prompt $\mathrm{Prompt}$, time series processed tokens and optional text tokens are fed into $f_{\mathrm{LLM}}^{\Delta}(\cdot)$ with partial unfreezing or additional adapter layers. $\mathcal{M}_{\mathrm{outp}}$ can be either a fine-tuned result or a intermediate result;

(iv) $\widehat{Y} = \mathrm{Task}(\mathcal{M}_{\mathrm{outp}})$: To generate or output required label $\widehat{Y}$, an extra task operation, denoted as $\mathrm{Task}(\cdot)$, is finally introduced to perform different analysis tasks.

## 3 METHODOLOGIES

### 3.1 ABBA SYMBOLIC APPROXIMATION

Our research is inspired by the observation that speech signals often contain a plethora of semantic information (van den Oord et al., 2016), which enables the language model to perform extremely well across a multitude of tasks; see Jin et al. (2024) and references therein. However, directly applying language models to time series is not permitted due to the fact that time series are made up of numerical values and lack useful embedding patterns; further, the high dimensionality of time series makes it difficult for the sequential and recurrent model to capture the dependency of the time series features. Thus learning an informative symbolic time series representation while having dimensionality reduced is a practical yet challenging problem. ABBA—a symbolic approximation method—is designed to address this as it compresses the time series to a symbolic presentation in terms of amplitude and period, and each symbol describes the ups and downs behavior of time series during a specific period. In the following, we will formulate the ABBA method that can be adapted to LLMs.

ABBA (Elsworth & Güttel, 2020) utilizes adaptive polygonal chain approximation followed by mean-based clustering to achieve symbolization of time series. The reconstruction error of the representation can be modeled as a *Brownian bridge* with pinned start and end points. ABBA symbolization contains two dominant procedures, namely *compression* and *digitization*, to aggregate time series $T = [t_1, t_2, \ldots, t_n] \in \mathbb{R}^n$ into its symbolic representation:

$$A = [a_1, a_2, \ldots, a_N],  \tag{1}$$

where $N \ll n$ and $a_i$ is an element in a specific letter set $\mathcal{L}$, which is referred to as a *dictionary* in the ABBA procedure.

### 3.1.1 COMPRESSION

The ABBA compression is performed to compute an adaptive piecewise linear continuous approximation (APCA) of $T$. The ABBA compression plays an critical role in dimensionality reduction in ABBA symbolic approximation—a user-specific tolerance, denoted by `tol`, is given to determine the degree of the reduction. The ABBA compression proceeds by adaptively selecting $N + 1$ indices $i_0 = 0 < i_1 < \cdots < i_N = n$ given a tolerance `tol` such that the time series $T$ is well approximated by a polygonal chain going through the points $(i_j, t_{i_j})$ for $j = 0, 1, \ldots, N$. This leads to a partition of $T$ into $N$ pieces $p_j = (\texttt{len}_j, \texttt{inc}_j)$ that represents cardinality and increment of $T_{i_{j-1}:i_j} = [t_{i_{j-1}}, t_{i_{j-1}+1}, \ldots, t_{i_j}]$, which is calculated by $\texttt{len}_j \in \mathbb{N} := i_j - i_{j-1} \geq 1$ and $\texttt{inc}_j \in \mathbb{R} := t_j - t_{j-1}$. As such, each

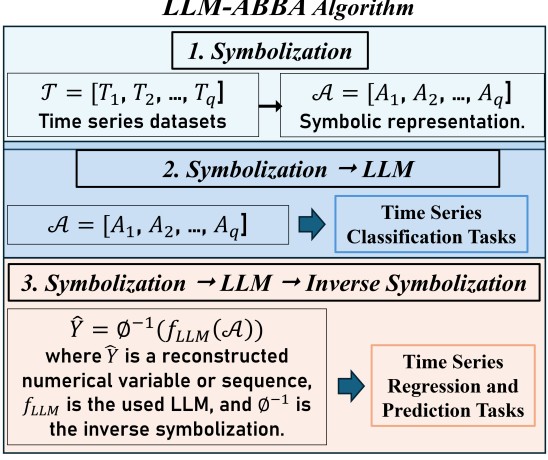

**LLM-ABBA Algorithm**

Figure 2: The framework of LLM-ABBA.

piece $p_j$ is represented by a straight line connecting the endpoint values $t_{i_{j-1}}$ and $t_{i_j}$. Given an index $i_{j-1}$ and starting with $i_0 = 0$, the procedure seeks the largest possible $i_j$ such that $i_{j-1} < i_j \leq n$ and

$$\sum_{i=i_{j-1}}^{i_j} \left( t_{i_{j-1}} + (t_{i_j} - t_{i_{j-1}}) \cdot \frac{i - i_{j-1}}{i_j - i_{j-1}} - t_i \right)^2 \leq (i_j - i_{j-1} - 1) \cdot \texttt{tol}^2.  \tag{2}$$

This means that this partitioning criterion indicates that the squared Euclidean distance of the values in $p_j$ from the straight polygonal line is upper bounded by $(\texttt{len}_j - 1) \cdot \texttt{tol}^2$.

Following the above, the whole polygonal chain can be recovered exactly from the first value $t_0$ and the tuple sequence $[p_1, p_2, \ldots, p_N]$ in the sense that the reconstruction error of this representation is with pinned start and end points and can be naturally modeled as a Brownian bridge. In terms of equation 2, a lower `tol` value is required to ensure an acceptable compression of time series with a great variety of features such as trends, seasonal and nonseasonal cycles, pulses and steps. As indicated in (Elsworth & Güttel, 2020), the error bound between the reconstruction and original time series is upper bounded by $(n - N) \cdot \texttt{tol}^2$.

### 3.1.2 DIGITIZATION

The ABBA compression is followed by a reasonable digitization that leads to a *symbolic representation* in the form of equation 1. Prior to digitizing, the tuple lengths and increments are separately normalized by their standard deviations $\sigma_{\texttt{len}}$ and $\sigma_{\texttt{inc}}$, respectively. After that, further scaling is employed by using a parameter `scl` to assign different weights to the length of each piece $p_i$, which denotes the importance assigned to its length value in relation to its increment value. Hence, the clustering is effectively performed on the *scaled tuples*

$$p_1' = \left( \texttt{scl} \frac{\texttt{len}_1}{\sigma_{\texttt{len}}}, \frac{\texttt{inc}_1}{\sigma_{\texttt{inc}}} \right), p_2' = \left( \texttt{scl} \frac{\texttt{len}_2}{\sigma_{\texttt{len}}}, \frac{\texttt{inc}_2}{\sigma_{\texttt{inc}}} \right), p_N' = \left( \texttt{scl} \frac{\texttt{len}_N}{\sigma_{\texttt{len}}}, \frac{\texttt{inc}_N}{\sigma_{\texttt{inc}}} \right).  \tag{3}$$

In particular, if $\texttt{scl} = 0$, then clustering will be only performed on the increment values of $p_i'$, while if $\texttt{scl} = 1$, the lengths and increments are treated with equal importance.

The step after normalization works with a mean-based clustering technique in Euclidean space. In the ABBA setting, letting the input of $N$ vectors be $P' = [p'_1, \ldots, p'_N] \in \mathrm{R}^{\ell \times N}$, one seeks a codebook of $k$ vectors, i.e., $C = [c_1, \ldots, c_k] \in \mathrm{R}^{\ell \times k}$ ($k \ll N$) where each $c_i$ is associated with a unique cluster $S_i$ such that $k$ clusters from $P'$ minimize the sum of Euclidean distances SSE constructed by $C$. The obtained codebook vectors are known as cluster centers. A quality codebook produces $k$ clusters $S_1, S_2, \ldots, S_k \subseteq P'$ such that the sum of squared errors SSE $= \sum_{i=1}^{k} \sum_{p' \in S_i} \|p' - c_i\|_2^2$ is small enough to an optimal level. To ensure SSE decreases as the iterations proceed, the mean value $\mu_i := \frac{1}{|S_i|} \sum_{p' \in S_i} p'$ is always chosen for centers update in $c_i$ for Lloyd's algorithm (Lloyd, 1982) (also known as the classic k-means algorithm). However, this is a suboptimal solution to minimizing SSE. The k-means problem aims to find $k$ clusters within data in $d$-dimensional space, so as to minimize the SSE. However, solving this problem is NP-hard even if $k$ is restricted to 2 (Drineas et al., 2004; Dasgupta & Freund, 2008) or in the plane (Mahajan et al., 2012). Typically, the sub-optimal k-means problem in the digitization can also be solved by a greedy sorting-based aggregation (Chen & Güttel, 2022). In the following, we assume the digitization is performed by the aggregation (See Algorithm 1 of Appendix) since this results in a faster variant, the number of symbols thus is determined by the parameter $\alpha$ (detail can be referred to Chen & Güttel (2022)).

In the context of symbolic approximation, we refer to the cluster centers as *symbolic centers* here, and each symbolic center is associated with an identical symbol. Then, each $p'_i$ is assigned to the closest symbolic center $c^i$ associated with its symbol $c^i = \arg\min_{c \in C}(\|p' - c\|)$. After that, each $p'_i$ is associated with a unique center, which is assigned as a label. We use a symbol to correspond to the label. The symbols can be represented by text characters, which are not limited to English alphabet letters—e.g., ASCII codes or any of its combinations.

### 3.1.3 Inverse symbolization

The *inverse symbolization* step converts the symbolic representation $A$ back to the reconstructed series $\widehat{T}$, which is key for some value prediction tasks in time series. The inverse symbolization is followed by a *inverse-digitization* that uses the $k$ representative elements $c_i \in C$ to replace the symbols in $A$ and denormalize them separately, thus resulting in a 2-by-$N$ array $\widetilde{P}$—an approximation of $P$. Each $\widetilde{p}_i \in \widetilde{P}$ is the closest symbolic center $c^i \in C$ to $p'_i \in P'$ (in contrast to $P$) after denormalization. However, the inverse digitization often leads to non-integer values for the reconstructed length len, so a rounding method is used to align the accumulated lengths with the closest integers. The first length is rounded to an integer value, i.e., $\widehat{\mathrm{len}}_1 := \mathrm{round}(\widetilde{\mathrm{len}}_1)$ and the rounding error $e := \widetilde{\mathrm{len}}_1 - \widehat{\mathrm{len}}_1$ is computed. The error is then added to the rounding of $\widetilde{\mathrm{len}}_2$, i.e., $\widehat{\mathrm{len}}_2 := \mathrm{round}(\widetilde{\mathrm{len}}_2 + e)$, and the new error $e'$ is calculated as $\widehat{\mathrm{len}}_2 + e - \widetilde{\mathrm{len}}_2$. Then $e'$ is similarly involved in the next rounding. After all rounding is computed, we obtain

$$\widehat{P} = [(\widehat{\mathrm{len}}_1, \widehat{\mathrm{inc}}_1), (\widehat{\mathrm{len}}_2, \widehat{\mathrm{inc}}_2), \ldots, (\widehat{\mathrm{len}}_N, \widehat{\mathrm{inc}}_N)] \in \mathbb{R}^{2 \times N}, \tag{4}$$

where the increments inc are unchanged, i.e., $\widehat{\mathrm{inc}} = \widetilde{\mathrm{inc}}$. The last step is to recover $\widehat{P}$ exactly from the initial time value $t_0$ and the tuple sequence equation 4, resulting in the reconstructed time series $\widehat{T}$.

## 3.2 ABBA to LLM

In the following, we write a single time series containing $n$ data points as $T$, and use $\mathcal{T} = \{T_i\}_{i=1}^{q}$ to denote a set of time series of cardinality $q$, associated with its corresponding symbolic representation set $\mathcal{A} = \{A_i\}_{i=1}^{q}$.

### 3.2.1 Symbolizing multiple time series

Existing work on symbolic approximation focuses converting a single time series; it can not convert another time series with consistent symbolic information (the same symbol correspond to the same symbolic center). To allow the manipulation of co-evolving time series or multiple time series, it is necessary to keep consistent symbolic information for multiple symbolic time series representations.

We illustrate a unified approach towards a consistent symbolic approximation to multiple time series.

- Step 1: Use APCA to compress each time series $T_i$ into $P_i$ for $i = 1, \ldots q$
- Step 2: Concateneate $P := [P_i]i = 1^q$

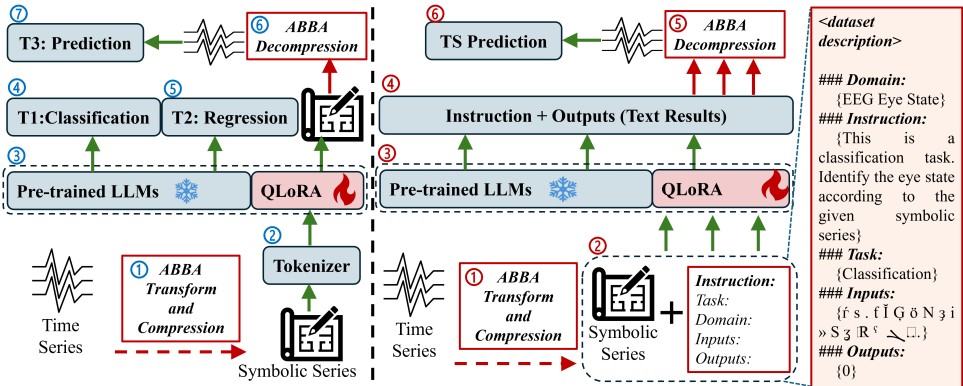

Figure 3: The model framework of LLM-ABBA.

- Step 3: Perform digization on $P$

- Step 4: Allocate symbols to each time series (the number of symbols for $T_i$ is equal to $|P_i|$)

### 3.2.2 SYMBOLIZING OUT-OF-SAMPLE DATA

Symbolizing the out-of-sample time series with consistent symbols is essential for various time series downstream tasks, which is used for inference tasks, etc. Given a set of time series $\mathcal{T} = \{T_i\}_{i=1}^q$, to compute a symbolic representation for $T_i(i = 1, \ldots m)$, we perform the following steps:

- Step 1: Compress each time series $T_i$ into $P_i$ for $i = 1, \ldots q$

- Step 2: Assign a symbol to $p \in P_i$ for $i = 1, \ldots q$ following the rule of digitization

### 3.2.3 FEEDING THE LLM

ABBA can transform numerical time series to symbolic series and keep the internal logic chain from which LLMs can learn the temporal knowledge. In other words, by ensuring the precondition that the input symbolic series inherits the polygonal chain of numerical time series and then represents this chain via symbolic series (or LLMs' tokens) that can be recognized by LLMs, LLMs can **reconstruct** the embedding space without the use of any new tokens via adapting fine-tuning methods. As seen in Figure 3, the left panel is the traditional setting in terms of corresponding tasks, such as classification, regression, and prediction. The right panel is the instruction setting that contains these three tasks. Given an input time series, we first transform and compress the time series to a symbolic series via ① and ①. These symbolic series will be tokenized by the LLM's tokenizer ②. The designed instruction that contains the symbolic series also will be tokenized by the LLM's tokenizer ②. Additionally, by only fine-tuning the pretrained LLM, the QLoRA with inhibition mechanism is utilized both in ③ and ③. To implement the corresponding tasks, ④ and ⑤ loads the LLM according to the type of task. However, ④ loads the LLM on the generation task. Moreover, to inverse symbolic series back to numerical time series, ⑥ and ⑤ utilizes ABBA to decompress the generated symbolic series. Lastly, in ⑦ and ⑥ the output time series from LLM-ABBA are projected to generate the forecasts.

For the consistency of the related tuning-based methods, $\mathcal{T}$ is referred to as the input in the time series dataset, $\mathcal{A}$ is the symbolic representation generated by ABBA; $\phi : \mathcal{T} \rightarrow \mathcal{A}$ denotes the symbolization of ABBA, and $\phi^{-1} : \mathcal{A} \rightarrow \mathcal{T}$ is referred to as the inverse symbolization of ABBA. We formulate the framework of LLM-ABBA:

(i) $\mathcal{A} = \phi(\mathcal{T})$: The input the $\mathcal{T}$ is converted to its symbolic representation $\mathcal{A}$.

(ii) $\mathcal{M}_{\text{inp}} = \text{Tokenizer}(\text{Prompt}, \mathcal{A})$: Tokenizing the symbolic representation $\mathcal{A}$; here, the Tokenizer is the default Tokenizer for LLMs.

(iii) $\mathcal{M}_{\text{outp}} = f_{\text{LLM}}^{\mathcal{A}}(\mathcal{M}_{\text{inp}})$: Feed the tokenized input to LLM model.

(iv) $\widehat{Y} = \mathrm{Task}\left(\mathcal{M}_{\mathrm{outp}}\right)$: If this is a classification task, $\widehat{Y}$ is a generated label. If the task is a regression or prediction task, $\widehat{Y}$ is an ABBA transformed numerical value or sequence produced by the inverse symbolization process of ABBA:

$$\begin{cases} \widehat{Y} = \mathcal{M}_{\mathrm{outp}}, & \text{Classification task,} \\ \widehat{Y} = \phi^{-1}\left(\mathcal{M}_{\mathrm{outp}}\right), & \text{Regression / prediction task} \end{cases}$$

### 3.3 FIXED-POINT ADAPTIVE POLYGONAL CHAIN

In time series prediction settings, the value-based prediction is converted into a token-based prediction using symbolic time series approximation. However, it is very desirable to mitigate the negative effect of the preceding mistakenly predicted symbol on the subsequent time series recovery since the recovery proceeds from front to back. However, APCA and the symbolic recovery often lead to a cumulative error for symbolic prediction, that is, a replacement of a previous symbol will influence the subsequent reconstruction. A *fixed-point polygonal chain* trick is introduced to mitigate this issue. We still partition the time series into pieces following equation 2 while $p_j = (\mathtt{len}_j, \mathtt{inc}_j)$ is replaced with $p_j = (\mathtt{len}_j, t_{i_j})$. We call the new approximation method FAPCA. The resulting tuples $p_i$ after normalization are equivalent to equation 3 and one can be recovered from the other since $\mathtt{inc}_j = t_{i_j} - t_{i_{j-1}}$. Figure 4 in the Appendix shows that FAPCA eliminates the cumulative errors arising from the preceding mistaken symbol and improves the recovery.

## 4 EXPERIMENTS

In this section, there are three time series tasks that will validate the efficiency of ABBA in LLM. We also fine-tune three language models on the training data using QLoRA (Dettmers et al., 2024) with inhibition (Kang et al., 2024). All experiments are simulated in PyTorch (Paszke et al., 2019) with a single NVIDIA A100 40GB GPU. The benefits of LLM-ABBA with LLMs include (1) avoiding the need for LLMs to learn time series from scratch, (2) only utilizing compression and decompression without the need for the training of extra embedding layers (Jin et al., 2023).

Table 1: Performance comparison of test accuracy (%) on 24 UCR time series classification datasets (Dau et al., 2019). Full results are shown in Table 9 of Appendix.

| Data Name | Classes Number | Symbols Number | M1 (%) | M2 (%) | M3 (%) | SOTA |
|---|---|---|---|---|---|---|
| BME | 3 | 836 | 60.2 | **84.7** | 77.3 | - |
| BeetleFly | 2 | 731 | **95.0** | 65.0 | 75.0 | - |
| ChinaTown | 2 | 585 | 72.6 | 84.3 | **89.2** | - |
| Coffee | 2 | 701 | 89.3 | 96.5 | 89.3 | **100** |
| ECG200 | 2 | 1,781 | 70.0 | 64.0 | 68.0 | **87.4** |
| ECG5000 | 5 | 10,334 | 81.2 | 76.0 | 75.4 | **94.0** |
| Earthquakes | 2 | 940 | 74.8 | 77.7 | **79.1** | 78.4 |
| GunPoint | 2 | 791 | 73.3 | 82.7 | 80.0 | **96.7** |
| GunPointAgeSpan | 2 | 2,057 | **94.3** | 84.5 | 85.5 | - |
| GunPointOldVersusYoung | 2 | 2,057 | **97.5** | 85.1 | 80.0 | - |
| HandOutlines | 2 | 7,572 | 77.0 | 68.6 | 71.6 | **93.2** |
| Herring | 2 | 982 | 65.6 | 62.5 | 60.9 | **68.8** |
| HouseTwenty | 2 | 1,385 | 86.2 | 89.1 | **93.3** | - |
| ItalyPowerDemand | 2 | 1,759 | 70.4 | 73.4 | 73.2 | **97.1** |
| Plane | 7 | 1,424 | 81.0 | 78.1 | **83.8** | - |
| PowerCons | 2 | 2,007 | 79.0 | **81.1** | 80.6 | - |
| SmallKitchenAppliances | 2 | 2,207 | 69.3 | 63.2 | 61.6 | **83.5** |
| Strawberry | 2 | 3,593 | 85.1 | 84.9 | 88.4 | **97.6** |
| Trace | 4 | 870 | 88.0 | 90.0 | 77.0 | **100** |
| TwoLeadECG | 2 | 2,487 | 69.1 | 64.6 | 63.9 | **97.8** |
| Wafer | 2 | 4,805 | 96.8 | 93.5 | 95.2 | **100** |
| Wine | 2 | 171 | 57.4 | 63.2 | 63.0 | **90.7** |
| Worms | 5 | 5,377 | 67.5 | 64.9 | 63.6 | **83.1** |
| WormsTwoClass | 2 | 5,377 | 81.8 | 70.1 | 79.2 | **98.7** |

### 4.1 COMPRESSION AND RECOVERY

To transform the numerical time series to symbolic time series, we merely use tokens of LLMs as the initial dictionary of ABBA for the symbolic representation, and there are no extra tokens that will be used to represent the numerical input. ABBA shows a strong symbolic transition on time series signals (See Figure 6 and Table 8 of the Appendix). Due to the page limitation, we report the performance of ABBA on time series transition in the Appendix.

### 4.2 PRETRAINED LARGE LANGUAGE MODELS

For a comprehensive analysis, we test ABBA with LLMs on three main time series analysis tasks. In this section, three LLMs are used to process the COP in symbolic series. M1 is the

RoBERTa$_{Large}$ (Liu et al., 2019), M2 means the Llama2-7B (Touvron et al., 2023), and M3 is the Mistral-7B (Jiang et al., 2023).

## 4.3 TIME SERIES CLASSIFICATION TASKS

For the classification task, we evaluate these three pretrained LLMs on UCR Time Series Archive datasets (Dau et al., 2019), EEG eye state (Seyfi et al., 2022), and MIT-BIH (Mousavi & Afghah, 2019; Liu & Zhang, 2021) which have been extensively adopted for benchmarking time series classification models. We utilise cross-entropy loss for the classification training. Details of the implementation and datasets can be found in Table 5 of the Appendix. The evaluation metric is accuracy rate (%).

In Table 1, we report the classification performance on a partial dataset of UCR2018. In most cases, although LLM-ABBA cannot outperform the SOTA in terms of time series classification tasks, ABBA with LLMs can reach an acceptable application requirement in some practical cases (such as 'Coffee', 'Earthquakes', 'Herring', 'Strawberry', 'Trace', 'Wafer', 'WormsT-woClass'). Because every UCR2018 data set is small and QLoRA has a relatively greater weight, we conclude that LLMs tend to suffer the over-fitting performance. Compared to V2S (Yang et al., 2021) which is the up-to-date SOTA, although these three LLMs with the use of QLoRA occupies more memory, the multi-modality of LLM especially on time series analysis requirement gets a noticeable improvement.

Table 2: Performance of test accuracy (%) on 3 medical time series classification datasets. Full results are shown in Table 10 of Appendix.

| Data | Classes Number | Symbols Number | M1 (%) | M2 (%) | M3 (%) | CNN (%) | BiRNN (%) | LSTM (%) |
|------|---------------|---------------|--------|--------|--------|---------|-----------|----------|
| EEG | 2 | 938 | 66.0 | 57.5 | 60.1 | * | * | |
| PTB-DB | 2 | 2179 | 90.6 | 99.0 | 98.9 | **99.4** | - | - |
| MIT-BIH | 5 | 2926 | 86.4 | 89.6 | 89.7 | 93.4 | **96.5** | 88.1 |

In the medical domain (for example, identifying the eye state using EEG signals, distinguishing the abnormal ECG signal, classifying the "normal beats", "supraventricular ectopy beats", "ventricular ectopy beats", "fusion beats" and "unclassifiable beats" of ECG signals), we report the performance of LLM-ABBAs on three medical time series datasets. We set tol = 0.01 and $\alpha$ = 0.01. In Table 2, Compared to CNN (Kachuee et al., 2018) in terms of PTB-DB data set, the LLM-ABBAs almost equalise the SOTA. In the aspect of distinguishing MIT-BIH, CNN (Kachuee et al., 2018) and BiRNN (Mousavi & Afghah, 2019) presents the dominant advantage, but LLM-ABBAs slightly outperforms LSTM (Singh et al., 2018).

Table 3: Performance comparison of the regression task (RMSE) on 19 Monashe Time Series Regression datasets. Our proposed LLM-ABBA outperforms or ties with the current prediction results on 19 out of 30 datasets. Full results are shown in Table 11 of Appendix.

| Data | Symbols Number | M1 (RMSE) | M2 (RMSE) | M3 (RMSE) | SOTA |
|------|---------------|-----------|-----------|-----------|------|
| AppliancesEnergy | 778 | **1.73** | 2.43 | 2.02 | 2.29 |
| HouseholdPowerConsumption1 | 1,717 | 377.02 | 398.01 | 228.67 | **132.80** |
| HouseholdPowerConsumption2 | 1,717 | 27.64 | 36.63 | **24.51** | 32.61 |
| BenzeneConcentration | 3,037 | 4.00 | 5.56 | 4.03 | **0.64** |
| BeijingPM10Quality | 970 | 66.07 | 93.25 | **65.24** | 93.14 |
| BeijingPM25Quality | 970 | 54.16 | 76.73 | **53.49** | 59.50 |
| LiveFuelMoistureContent | 5,689 | **20.56** | 29.32 | 20.85 | 29.41 |
| FloodModeling1 | 969 | 0.00 | 0.05 | 0.36 | **0.00** |
| FloodModeling2 | 979 | **0.00** | 0.04 | 0.39 | 0.01 |
| FloodModeling3 | 948 | 0.00 | 0.05 | 0.37 | **0.00** |
| AustraliaRainfall | 4,740 | 4.36 | 6.01 | **4.28** | 8.12 |
| PPGDalia | 12,298 | 9.32 | 12.50 | **9.02** | 9.92 |
| IEEEPPG | 8,971 | **17.00** | 22.53 | 17.12 | 23.90 |
| BIDMC32HR | 9,423 | **6.98** | 11.98 | 8.21 | 9.42 |
| BIDMC32RR | 9,412 | **1.74** | 2.61 | 2.06 | 3.02 |
| BIDMC32SpO2 | 5,537 | **2.85** | 3.79 | 2.91 | 4.45 |
| NewsHeadlineSentiment | 5,537 | **0.07** | 0.13 | 0.11 | 0.14 |
| NewsTitleSentiment | 5,537 | **0.07** | 0.13 | 0.11 | 0.14 |
| Covid3Month | 227 | **0.02** | 0.11 | 0.44 | 0.04 |

## 4.4 TIME SERIES REGRESSION TASKS

For the regression task, we evaluate these three pretrained LLMs on the Time Series Extrinsic Regression (TSER) benchmarking archive (Tan et al., 2021), which contains 19 time series datasets

Table 4: Performance comparison of the prediction task (MSE and MAE) on 4 time series prediction datasets. Full results are shown in Table 12 of the Appendix.

| Data | Predictor Length | Symbols Number | M2 | | M3 | | Informer | | UniTime | | Time-LLM | | AutoTimes | |
|---|---|---|---|---|---|---|---|---|---|---|---|---|---|---|
| | | | MSE | MAE | MSE | MAE | MSE | MAE | MSE | MAE | MSE | MAE | MSE | MAE |
| ETTh1 | 168/24 | 2789 | 0.653 | 0.647 | 0.622 | 0.631 | 0.577 | 0.549 | - | - | - | - | - | - |
| ETTh2 | 168/24 | 5383 | 0.784 | 0.761 | 0.759 | 0.761 | 0.720 | 0.665 | - | - | - | - | - | - |
| ETTm1 | 168/24 | 3170 | 0.386 | 0.364 | 0.401 | 0.387 | 0.323 | 0.369 | - | - | - | - | - | - |
| ETTm2 | 168/24 | 6878 | 0.201 | 0.198 | 0.214 | 0.203 | - | - | - | - | - | - | - | - |
| ETTh1 | 168/96 | 2789 | 0.745 | 0.752 | 0.773 | 0.782 | - | - | 0.397 | 0.418 | 0.362 | 0.392 | 0.360 | 0.400 |
| ETTh2 | 168/96 | 5383 | 0.892 | 0.881 | 0.871 | 0.866 | - | - | 0.296 | 0.345 | 0.268 | 0.328 | - | - |
| ETTm1 | 168/96 | 3170 | 0.531 | 0.528 | 0.524 | 0.517 | - | - | 0.322 | 0.363 | 0.272 | 0.334 | - | - |
| ETTm2 | 168/96 | 6878 | 0.288 | 0.267 | 0.276 | 0.281 | - | - | 0.183 | 0.266 | 0.161 | 0.253 | - | - |
| ETTh1 | 168/168 | 2789 | 1.087 | 0.964 | 1.174 | 1.968 | 0.931 | 0.752 | - | - | - | - | - | - |
| ETTh2 | 168/168 | 5383 | 3.975 | 2.101 | 3.898 | 2.134 | 3.489 | 1.515 | - | - | - | - | - | - |
| ETTm1 | 168/168 | 3170 | 0.974 | 0.952 | 0.966 | 0.958 | 0.678 | 0.614 | - | - | - | - | - | - |
| ETTm2 | 168/168 | 6878 | 0.576 | 0.544 | 0.521 | 0.503 | - | - | - | - | - | - | - | - |

from 5 application areas, including Health Monitoring, Energy Monitoring, Environment Monitoring, Sentiment Analysis and Forecasting[1]. To use as few symbols as possible, we initialize the setting of $tol = 0.01$ and $= 0.01$. We also utilize the L2 loss for the regression training. Details of the implementation and datasets can be found in Table 6 of the Appendix. The evaluation metric is root-mean-square-error (RMSE).

Experimenting on the TSER benchmark archive (Tan et al., 2021), the empirical results are shown in Table 3, in which 15 out of 19 use-cases outperform the machine learning SOTA results. We believe that LLM-ABBA can exploit the semantic information hiding beneath the time series in the task of time series regression. ABBA is able to provide COPs to LLMs by compressing and digitizing time series to symbols, which finally results in the change of embedding space by using adaption fine-tuning methods.

### 4.5 TIME SERIES FORECASTING TASKS

For time series forecasting, we experimented on 4 well-established benchmarks: ETT datasets (including 4 subsets: ETTh1, ETTh2, ETTm1, ETTm2) (Zhou et al., 2021; Wu et al., 2021). Details of the implementation and datasets can be found in Table 7 of Appendix. The input length of the time series is $168$, and we use three different prediction horizons $H \in \{24, 96, 168\}$. The evaluation metrics include mean square error (MSE) and mean absolute error (MAE).

Although LLM-ABBA cannot obtain a new SOTA on time series forecasting tasks, it compares favorably to the Informer architecture which is trained from scratch. The congenital defect of ABBA is that the symbolization tends to be affected by the fluctuation and oscillation of time series signals, which eventually leads to higher MSE and MAE scores. Because LLM-ABBA utilizes a totally different technical roadmap to existing methods, it only remolds the construction of the LLM's tokens. However, remodeling pretrained tokens inevitably brings the previous pretrained semantics to the LLM-ABBA design. Thus, we discussed the semantic consistency of LLM-ABBA using extra symbols or tokens to overcome this problem (See in Appendix).

### 5 CONCLUSION

In this paper, we propose LLM-ABBA for time series classification, regression, and forecasting tasks. We discuss how to seamlessly integrate time series symbolization with LLMs and enhance its performance. To mitigate the drift phenomenon of time series, we introduce the FAPCA method to improve ABBA symbolization. The empirical results demonstrate our method achieves the comparable SOTA performance on classification and regression tasks. We refer readers of interest to the Appendix for further discussion on the reconstruction error of ABBA symbolization, how it relates to the dominant parameters, and the congenital defect of LLM-ABBA. In terms of convenience and universality, LLM-ABBA improves the multi-modality of LLMs on time series analysis. We believe the potential of ABBA extends to other time series applications, which will be left as future work.

---

[1]Monash regression data is available at http://tseregression.org/.

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

## A  SORTING-BASED AGGREGATION

---

**Algorithm 1:** Fast sorting-based aggregation

---

1. Scale and sort data points, and assume they are denoted $p_1, \ldots, p_n$.
   Label all of them as "unassigned".
2. For $i \in \{1, \ldots, n\}$ let $p_i$ be the first unassigned point and set $j := i$.
   (The point $p_i$ is the *starting point* of a new group.)
   If there are no unassigned points left, go to Step 6.
3. Compute $d_{ij} := \mathrm{d}(p_i, p_j)$
4. If $d_{ij} \leq \alpha$,
   - assign $p_j$ to the same group as $p_i$
   - increase $j := j + 1$
5. If $j > n$ or termination condition is satisfied, go to Step 2. Otherwise go to Step 3.
6. For each computed group, compute the group center as the mean of all its points.

---

## B  ERROR ANALYSIS OF TIME SERIES RECONSTRUCTION

In this section, we are concerned with the reconstruction error of ABBA. It's note that the reconstruction of time series from compression procedure proceeds by establishing a polygonal chain $\widetilde{T}$ going through the chosen tuples $\{(i_j, t_{i_j})\}_{j=0}^{N}$ from the original time series $T$ and $len_j = i_{j+1} - i_j$. As indicated in (Elsworth & Güttel, 2020), a polygonal chain $\widehat{T}$ stitching together $\{(\widehat{i}_j, \widehat{t}_{i_j})\}_{j=0}^{N}$ via a tuple sequence $\widehat{P}$ is reconstructed by the inverse symbolization.

**Theorem B.1** ((Elsworth & Güttel, 2020)). *Let* $(\mu_i^{len}, \mu_i^{inc}) = \frac{1}{|S_i|} \sum_{(len, inc) \in S_i} (len, inc)$, *we denote the mean set for* len *and* inc *by* $\mathcal{U}_{len} = \{\mu_i^{len}\}_{i=1}^{k}$ *and* $\mathcal{U}_{inc} = \{\mu_i^{inc}\}_{i=1}^{k}$, *respectively. Due to* $i_0 = 0$, *the reconstruction indices and size of time series values are given by*

$$(\widehat{i}_j, \widehat{t}_{i_j}) = \left( \sum_{\ell=1}^{j} \widehat{len}_\ell, t_0 + \sum_{\ell=1}^{j} \widehat{inc}_\ell \right), \quad for \; j = 0, \ldots, N, \tag{5}$$

*where* $(\widehat{len}_\ell, \widehat{inc}_\ell)$ *computed cluster centers, i.e.,* $\widehat{len}_\ell \in \mathcal{U}_{len}$ *and* $\widehat{inc}_\ell \in \mathcal{U}_{inc}$.

Theorem B.1 shows the accumulated deviations from the true lengths and increments are canceled out (as analyzed in (Elsworth & Güttel, 2020)) at the right endpoint of the last piece $p_N$, thus $(\widehat{i}_N, \widehat{t}_{i_N}) = (i_N, t_{i_N}) = (n, t_n)$, which indicates the start and ending point between $\widehat{T}, \widetilde{T}$ and $T$ are identical. We thus have the following result.

Now we denote the local deviation of the increment and length:

$$d_\ell^{inc} := \widehat{inc}_\ell - \widetilde{inc}_\ell, \quad d_\ell^{len} := \widehat{len}_\ell - \widetilde{len}_\ell. \tag{6}$$

**Theorem B.2** ((Elsworth & Güttel, 2020)).

$$\sum_i \sum_{(len, inc) \in S_i} (d^{len}, d^{inc}) = (0, 0). \tag{}$$

**Theorem B.3.** *Consider ABBA is performed with hyperparameter* $\alpha$ *and result in* $k$ *clusters* $S_1, \ldots, S_k$, *then we have*

$$\max_\ell \{(d_\ell^{inc})^2 + (d_\ell^{len})^2\} \leq \alpha^2, \tag{7}$$

*and further*

$$\sigma = \max_{i=1,\ldots,k} \frac{1}{|S_i|} \sum_{(len, inc) \in S_i} \left( |len - \mu_i^{len}|^2 + |inc - \mu_i^{inc}|^2 \right) \leq \alpha^2, \tag{8}$$

Following Theorem B.3, the $\sigma$ is explicitly controlled by $\alpha$, thus we remove the need to estimate an additional parameter of $\mathtt{tol}_s$ stated in (Elsworth & Güttel, 2020) by directly relating it to hyperparameter $\alpha$.

Given the $N$ data points selected by adaptive polygonal approximation chain, let $e_j^{\text{len}} := \sum_{\ell=1}^{j} d_\ell^{\text{len}}$ and $e_j^{\text{inc}} := \sum_{\ell=1}^{j} d_\ell^{\text{inc}}$, it is obvious that $e_j^{\text{inc}} = \widehat{t}_{i_j} - t_{i_j}$ if $e_j^{\text{len}} = 0$ for $j = 1, \ldots, N$, then we have the Theorem B.4 and Theorem B.5.

**Theorem B.4.**

$$|e_j^{inc}| \leq j\sqrt{\alpha^2 - (d_\ell^{len})^2} \leq j|\alpha|, \tag{9}$$

*where $j = 0, \ldots, N$.*

*Similarly, the shift of the time series has $|e_j^{len}| \leq j\sqrt{\alpha^2 - (d_\ell^{inc})^2} \leq j|\alpha|$ for $j = 0, \ldots, N$.*

**Theorem B.5.**

$$\mathbb{P}(|e_j^{inc}| \geq h) \leq \exp\left(-\frac{h^2}{2j\alpha^2}\right) \quad and \quad \mathbb{P}(|e_j^{len}| \geq h) \leq \exp\left(-\frac{h^2}{2j\alpha^2}\right).$$

*for all $h > 0$.*

*Proof to Theorem B.5.* In terms of Theorem B.2, we can easily obtain

$$(e_0^{\text{len}}, e_0^{\text{inc}}) = (0,0), \quad (e_N^{\text{len}}, e_N^{\text{inc}}) = (0,0)$$

associated with expectation $E(e_j^{\text{len}}) = E(e_j^{\text{len}}) = 0$.

For $j = 1, \ldots, N$, since $d_j^{\text{len}}, d_j^{\text{inc}} \in [-\alpha, \alpha]$, in terms of equation 7 and Hoeffding's inequality, then

$$\mathbb{P}\left(\left|\sum_{\ell=1}^{j}(d_\ell^{\text{inc}} - E[d_\ell^{\text{inc}}])\right| \geq h\right) = \mathbb{P}\left(\left|e_j^{\text{inc}} - E[e_j^{\text{inc}}]\right| \geq h\right) \leq \exp\left(-\frac{h^2}{2j\alpha^2}\right) \tag{10}$$

Therefore,

$$\mathbb{P}(|e_j^{\text{len}}| \geq h) \leq \exp\left(-\frac{h^2}{2j\alpha^2}\right) \quad \text{and} \quad \mathbb{P}(|e_j^{\text{inc}}| \geq h) \leq \exp\left(-\frac{h^2}{2j\alpha^2}\right).$$

for all $t > 0$. $\qquad\square$

Therefore, a decrease of $\alpha$ is prone to result in a smaller reconstruction error $e_j$, this phenomenon was mentioned in (Elsworth & Güttel, 2020). The growth of $j$ increases the possibility of larger errors since the errors coming from the previous reconstruction will be accumulated to the subsequent reconstruction in terms of the principle of inverse symbolization.

## C  LINGUISTICS INVESTIGATION: ZIPF'S LAW

The most common word is often found to appear approximately twice as frequently as the next common word, this phenomenon is explained by Zipf's law (Powers, 1998). Zipf's law asserts that the frequencies of certain events are inversely proportional to their rank, further, the rank-frequency distribution is an inverse power law relation.

Also, as depicted in Figure 5, we can see unigrams generated by ABBA symbolization from 7 different time series datasets of UCR Archive coarsely meet Zipf's law. This showcases an appealing alignment between ABBA symbols and the native language words.

## D  HYPERPARAMETERS

### D.1  HYPERPARAMETERS OF ABBA

There are four interactive parameters that establish the transition of time series when integrating ABBA to LLMs. $\texttt{tol} \in \{1e-2, 1e-4, 1e-6\}$ is the tolerance that is set to control the degree of the reduction, $\alpha \in \{1e-2, 1e-4, 1e-6\}$ sets the upper bound, $\mathcal{L}$ is a finite letter set that can be the LLMs' tokens, $init \in \{\text{'agg', 'k-means'}\}$ is the initial clustering method, and $scl \in \{1, 2, 3\}$ is used to weight the length of each piece.

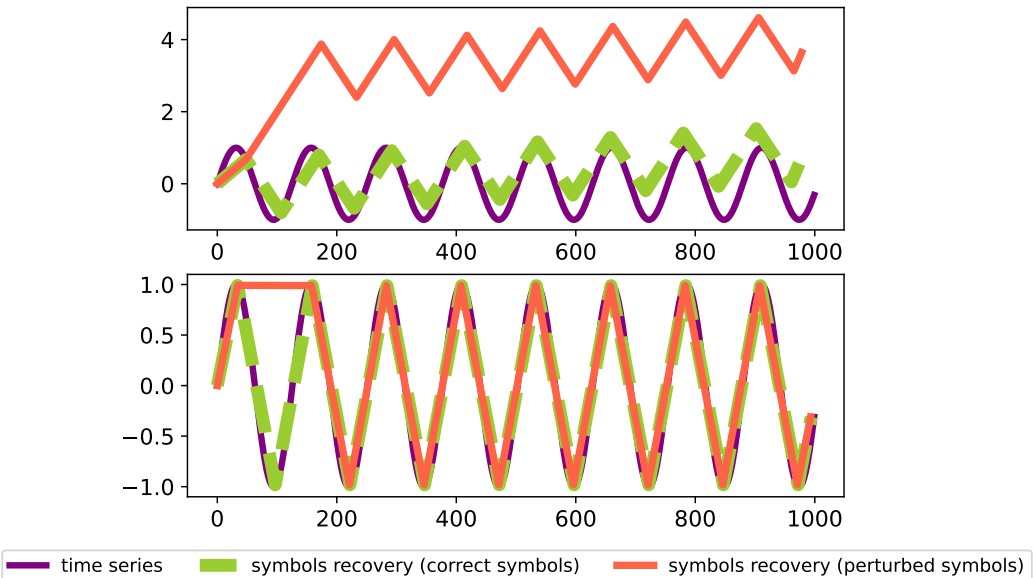

Figure 4: We generate synthetic trigonometric sine series of 1,000 points, and separately perform symbolic approximation with 4 symbols using APCA (top) and FAPCA (bottom) on the time series, respectively. APCA and FAPCA generate symbols 'aBbBbBbBbBbBbBbBbBA' and 'abBbBbBbBbBbBbBbBbA', respetively, associated with their perturbed symbols, 'abbBbBbBbBbBbBbBbBA' and 'aBBbBbBbBbBbBbBbBbA', correspondingly. The symbols recovery is performed on correct symbols and perturbed symbols, respectively.

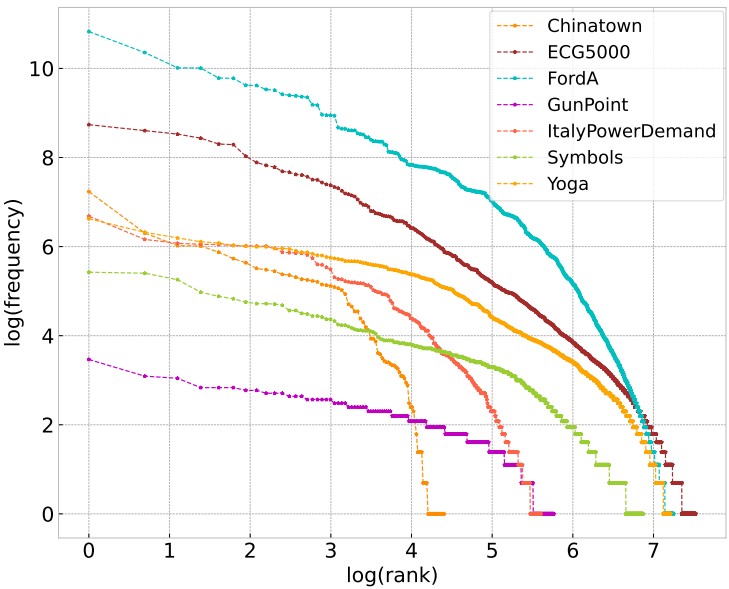

Figure 5: Frequency and rank of symbols in various UCR datasets.

## D.2 HYPERPARAMETERS OF LLMS

There are three time series analysis tasks: classification, regression and prediction. We quantize LLMs by 4-bits using the bitsandbytes package[2]. In order to fine-tune LLMs as accordingly as possible, the shunting inhibition mechanism (Kang et al., 2024) is utilized during the QLoRA

---

[2]https://github.com/bitsandbytes-foundation/bitsandbytes

Table 5: Hyperparameters of Classification tasks. Quant. is the model quantization process. Inhib. is the inhibition threshold in QLoRA. Embed. means to save tuned embeddings. Optims. is the optimization method. LR is the learning rate. Acc. is the accyracy rate (%).

| | Quant. | Tokens | Metric | LoRA | | | | | | Optim. | Epochs | LR | Batch |
| Models | 4-bits | Length | | alpha | low rank r | dropout | inhib. | Embed. | | | | Size |
|---|---|---|---|---|---|---|---|---|---|---|---|---|
| **LLM-ABBA on Classification Tasks** | | | | | | | | | | | | |
| RoBERTa$_{Large}$ | True | 512 | Acc. | 16 | 16, 64, 256 | 0.05 | 0.3 | Save | adamw_8bit | 10 | 5e-7 | 4 |
| Llama2-7B | True | 4096 | Acc. | 16 | 16, 64, 256 | 0.05 | 0.3 | Save | adamw_8bit | 10 | 5e-7 | 4 |
| Mistral-7B | True | 4096 | Acc. | 16 | 16, 64, 256 | 0.05 | 0.3 | Save | adamw_8bit | 10 | 5e-7 | 4 |

Table 6: Hyperparameters of Regression tasks. Quant. is the model quantization process. Inhib. is the inhibition threshold in QLoRA. Embed. means to save tuned embeddings. Optims. is the optimization method. RMSE is the root-mean-square-error.

| | Quant. | Tokens | Metric | LoRA | | | | | | Optim. | Epochs | LR | Batch |
| Models | 4-bit | Length | | alpha | low rank r | dropout | inhib. | Embed. | | | | Size |
|---|---|---|---|---|---|---|---|---|---|---|---|---|
| **LLM-ABBA on Regression Tasks** | | | | | | | | | | | | |
| RoBERTa$_{Large}$ | True | 512 | RMSE | 16 | 16, 64, 256 | 0.05 | 0.3 | Save | adamw_8bit | 10 | 2e-6 | 4 |
| Llama2-7B | True | 4096 | RMSE | 16 | 16, 64, 256 | 0.05 | 0.3 | Save | adamw_8bit | 10 | 2e-6 | 4 |
| Mistral-7B | True | 4096 | RMSE | 16 | 16, 64, 256 | 0.05 | 0.3 | Save | adamw_8bit | 10 | 2e-4 | 4 |

Table 7: Hyperparameters of Prediction tasks. Quant. is the model quantization process. Inhib. is the inhibition threshold in QLoRA. Embed. means to save tuned embeddings. Optims. is the optimization method. MAE is the mean-absolute-error, and MSE is the mean-square-error.

| | Quant. | Tokens | Metric | LoRA | | | | | | Optim. | Epochs | LR | Batch |
| Models | 4-bit | Length | | alpha | low rank r | dropout | inhib. | Embed. | | | | Size |
|---|---|---|---|---|---|---|---|---|---|---|---|---|
| **LLM-ABBA on Prediction Tasks** | | | | | | | | | | | | |
| RoBERTa$_{Large}$ | True | 512 | MAE, MSE | 16 | 16, 64, 256 | 0.05 | 0.3 | Save | adamw_8bit | 10 | 2e-6 | 4 |
| Llama2-7B | True | 4096 | MAE, MSE | 16 | 16, 64, 256 | 0.05 | 0.3 | Save | adamw_8bit | 10 | 2e-6 | 4 |
| Mistral-7B | True | 4096 | MAE, MSE | 16 | 16, 64, 256 | 0.05 | 0.3 | Save | adamw_8bit | 10 | 2e-6 | 4 |

(Dettmers et al., 2024) adapter fine-tuning progress. The modified embedding layer is also saved after fine-tuning on the corresponding task. For the classification task, the metric is accuracy rate (%). Root-mean-square-error is used to be the metric of regressing tasks. Mean-square-error and mean-absolute-error are the metrics of prediction tasks, and we also visualize the correlation coefficient of prediction tasks on ETTh1 data in terms of their seven features. We control the fine-tuning epoch and apply a small batch size on on every task. The alpha of QLoRA is set to 16. Every task is run and tested on a single 40G GPU.

# E  THE PERFORMANCE OF ABBA ON TIME SERIES TRANSITION

To visualize the performance of ABBA on time series transition process, we employ ETTh1 time series data to compute the correlation n coefficient and reconstruction error of ABBA. This multivariate data has seven features, and in terms of these seven features, the average of MSE, MAE and correlation coefficient between original time series input and reconstructed outputs is computed.

In this session, the default $scl$ is set to 3, and $init$ is 'agg' which is used in other LLM tasks. Meanwhile, tol and $\alpha$ are set to be the same. Table 8 reports the input-168-predict-96 results when using ABBA to reconstruct ETTh1 data in terms of seven features. Setting smaller tol and $\alpha$ of ABBA can reduce MSE and MAE scores, but more symbols or LLM's tokens will be used. Under all above conditions, the correlation coefficient is 1.0.

Table 8: Symbolic approximation performance on ETTh1 data by using ABBA.

| ABBA Settings | | Number of Symbols | Reconstructed Time Series | | |
|---|---|---|---|---|---|
| $tol$ and $alpha$ | $scl$ | Used LLM's tokens | MSE | MAE | Correlation Coefficient |
| 1e-2, 1e-2 | 3 | 846 | 2.5e-7 | 1.0e-2 | 1.0 |
| 1e-4, 1e-4 | 3 | 2713 | 4.2e-8 | 1.4e-4 | 1.0 |
| 1e-6, 1e-6 | 3 | 2789 | 3.2e-8 | 1.3e-4 | 1.0 |

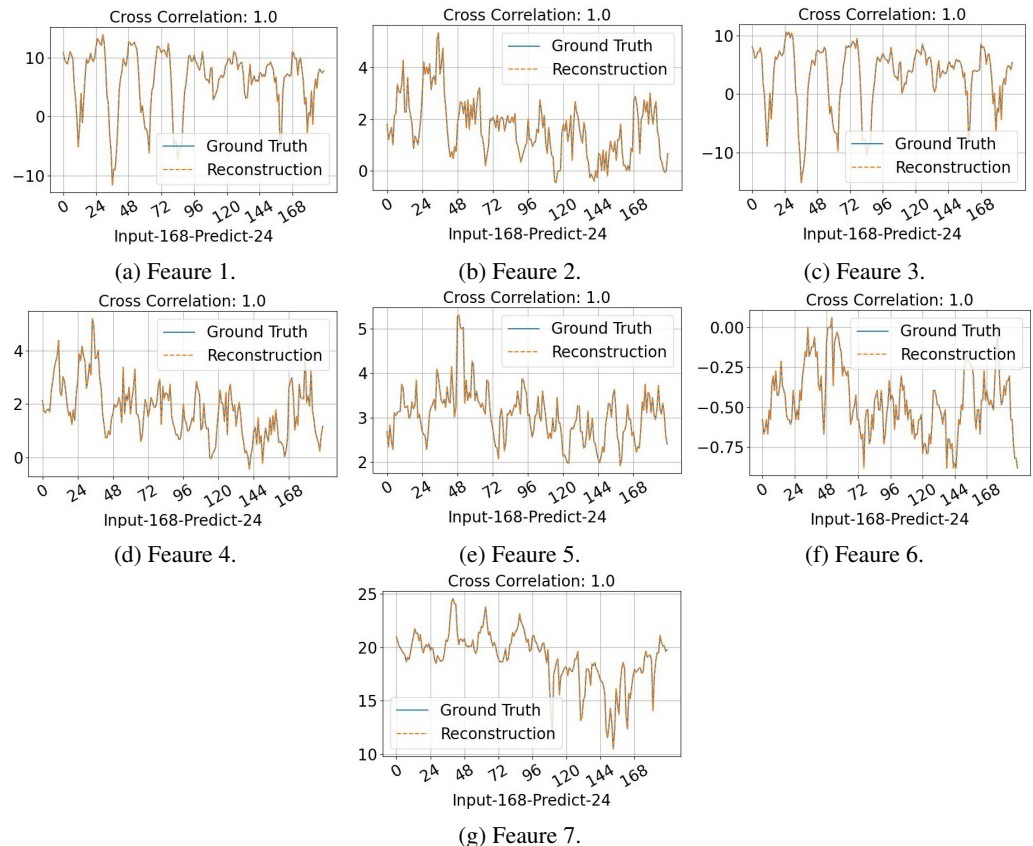

(a) Feaure 1.  (b) Feaure 2.  (c) Feaure 3.

(d) Feaure 4.  (e) Feaure 5.  (f) Feaure 6.

(g) Feaure 7.

Figure 6: Visualization of reconstructed input-168-predict-24 results on ETTh1 data by using ABBA symbolic approximation.

## F    FULL RESULTS

To verify that if LLM-ABBA has the over-fitting problem, we use different low rank of QLoRA on the corresponding tasks during the fine-tuning progress. Due to the small size of each data on UCR time series classification datasets, we set the $r$ of QLoRA to 16. But for time series regression and prediction tasks, we select $r \in \{16, 46, 256\}$ for the corresponding data input. We find that there is no obvious over-fitting problem, and more tunable parameters are not able to make LLM-ABBA performs better.

The UCR Archive contains 128 datasets has already been partitioned into train and test set while the ratio of the train set and test set is not always consistent in various datasets. These datasets have varying numbers of labels and feature dimensions, Also, might exist uneven numbers of labels which is very likely to result in the overfitting arise from imbalanced data problems. Therefore, classifying time series in UCR Archive is a challenging task. Table 9 reports the full time series classification results on UCR2018. J1 is the "k-means" symbolization method, and J2 is the "aggregation" symbolization. We find that "aggregation" outperforms "k-means" symbolization time series transition progress in most cases. A larger data needs more symbols or LLM's tokens, as a larger time series data would contain more information and symbolic semantics. RoBERTa$_{large}$ is based on BERT (Devlin, 2018) which considers two directions of the input language sentence, meanwhile, Llama2-7B and Mistral-7B originates from the GPT architecture (Radford et al., 2019) that only takes one direction (from left to right) into account. Causality analysis which should compute the contextual of each signal has been widely used to analyze multichannel EEG signals. However, ECG signals mostly rely on the sequential features. Thus, we infer that when using LLM-ABBA to analyze medical time series, the properties and characteristics should be analyzed first.

Table 9: Full comparison results of time series classification tasks(%) on UCR datasets.

| Data | Classes Number | Symbols Number | RoBERTa$_{Large}$ Para. | J1 | J2 | Llama2-7B Para. | J1 | J2 | Mistreal-7B Para. | J1 | J2 | SOTA Para. | V2Sa |
|---|---|---|---|---|---|---|---|---|---|---|---|---|---|
| BME | 3 | 836 | 2.65M | 34.0 | 60.2 | 12.7M | 41.3 | 84.7 | 9.56M | 43.3 | 77.3 | 0.3M | - |
| BeetleFly | 2 | 731 | 2.65M | 65.0 | 95.0 | 12.7M | 50.0 | 65.0 | 9.56M | 55.0 | 75.0 | 0.3M | - |
| BirdChicken | 2 | 424 | 2.65M | 55.0 | 70.0 | 12.7M | 60.0 | 65.0 | 9.56M | 55.0 | 75.0 | 0.3M | - |
| ChinaTown | 2 | 585 | 2.65M | 72.0 | 72.6 | 12.7M | 58.3 | 84.3 | 9.56M | 61.5 | 89.2 | 0.3M | - |
| Coffee | 2 | 701 | 2.65M | 50.0 | 89.3 | 12.7M | 60.7 | 96.5 | 9.56M | 78.6 | 89.3 | | 100 |
| Computers | 2 | 2,587 | 2.65M | 37.4 | 70.8 | 12.7M | 65.8 | 60.4 | 9.56M | 63.2 | 64.4 | 0.3M | - |
| DiatomSizeReduction | 4 | 940 | 2.65M | 38.6 | 52.3 | 12.7M | 33.7 | 54.3 | 9.56M | 36.3 | 52.0 | 0.3M | - |
| DistalPhalanxOutlineCorrect | 2 | 2,125 | 2.65M | 58.3 | 63.8 | 12.7M | 62.0 | 68.8 | 9.56M | 58.3 | 66.7 | 0.3M | - |
| DistalPhalanxTW | 6 | 1,444 | 2.65M | 58.3 | 61.2 | 12.7M | 64.0 | 59.7 | 9.56M | 56.8 | 62.6 | | 79.1 |
| DodgerLoopWeekend | 2 | 143 | 2.65M | 72.6 | 73.9 | 12.7M | 70.3 | 64.5 | 9.56M | 69.6 | 71.7 | 0.3M | - |
| ECG200 | 2 | 1,781 | 2.65M | 70.0 | 68.0 | 12.7M | 63.0 | 64.0 | 9.56M | 66.8 | 68.0 | | 87.4 |
| ECG5000 | 5 | 10,334 | 2.65M | 81.2 | 76.0 | 12.7M | 75.7 | 74.7 | 9.56M | 75.4 | 73.4 | 0.3M | 94.0 |
| ECGFiveDays | 2 | 2,463 | 2.65M | 52.6 | 56.9 | 12.7M | 53.3 | 63.9 | 9.56M | 49.5 | 68.8 | 0.3M | - |
| Earthquakes | 2 | 940 | 2.65M | 52.7 | 74.8 | 12.7M | 77.7 | 76.3 | 9.56M | 79.1 | 76.3 | 0.3M | 78.4 |
| ElectricDevices | 7 | 4,607 | 2.65M | 34.2 | 56.6 | 12.7M | 54.9 | 51.0 | 9.56M | 52.7 | 51.6 | 0.3M | - |
| FordA | 2 | 9,759 | 2.65M | 68.9 | 68.9 | 12.7M | 58.7 | 61.1 | 9.56M | 62.7 | 60.9 | 0.3M | 100 |
| FordB | 2 | 9,352 | 2.65M | 68.9 | 58.1 | 12.7M | 56.1 | 58.9 | 9.56M | 55.1 | 57.0 | 0.3M | 100 |
| FreezerRegularTrain | 2 | 2,663 | 2.65M | 61.9 | 74.5 | 12.7M | 64.1 | 76.1 | 9.56M | 63.2 | 75.4 | 0.3M | - |
| FreezerSmallTrain | 2 | 2,593 | 2.65M | 62.3 | 74.1 | 12.7M | 63.8 | 67.8 | 9.56M | 63.3 | 67.5 | 0.3M | - |
| GunPoint | 2 | 791 | 2.65M | 51.4 | 73.3 | 12.7M | 54.0 | 82.7 | 9.56M | 48.0 | 80.0 | 0.3M | 96.7 |
| GunPointAgeSpan | 2 | 2,057 | 2.65M | 83.5 | 94.3 | 12.7M | 69.9 | 84.5 | 9.56M | 67.1 | 85.5 | 0.3M | - |
| GunPointMaleVersusFemale | 2 | 2,057 | 2.65M | 57.9 | 76.3 | 12.7M | 59.8 | 71.2 | 9.56M | 55.7 | 74.1 | 0.3M | - |
| GunPointOldVersusYoung | 2 | 2,057 | 2.65M | 66.7 | 97.5 | 12.7M | 62.9 | 85.1 | 9.56M | 67.9 | 80.0 | 0.3M | - |
| HandOutlines | 2 | 7,572 | 2.65M | 66.5 | 77.0 | 12.7M | 63.5 | 68.6 | 9.56M | 65.1 | 71.6 | 0.3M | 93.2 |
| Herring | 2 | 982 | 2.65M | 59.4 | 65.6 | 12.7M | 62.5 | 62.5 | 9.56M | 54.7 | 60.9 | 0.3M | 68.8 |
| HouseTwenty | 2 | 1,385 | 2.65M | 50.8 | 67.1 | 12.7M | 69.7 | 89.1 | 9.56M | 75.6 | 93.3 | 0.3M | - |
| ItalyPowerDemand | 2 | 1,759 | 2.65M | 59.7 | 70.4 | 12.7M | 55.7 | 73.4 | 9.56M | 53.4 | 73.2 | 0.3M | 97.1 |
| LargeKitchenAppliances | 2 | 3,067 | 2.65M | 39.6 | 63.5 | 12.7M | 46.4 | 64.1 | 9.56M | 42.1 | 54.7 | 0.3M | - |
| Lightning2 | 2 | 2,175 | 2.65M | 67.2 | 65.6 | 12.7M | 68.9 | 65.6 | 9.56M | 67.2 | 62.3 | 0.3M | 100 |
| Meat | 3 | 161 | 2.65M | 55.0 | 70.0 | 12.7M | 68.3 | 70.0 | 9.56M | 66.7 | 70.0 | 0.3M | - |
| MedicalImages | 10 | 4,173 | 2.65M | 52.5 | 51.8 | 12.7M | 49.2 | 49.9 | 9.56M | 48.2 | 49.5 | 0.3M | - |
| MelbournePedestrian | 10 | 1,081 | 2.65M | 34.6 | 68.5 | 12.7M | 27.1 | 76.8 | 9.56M | 29.2 | 74.4 | 0.3M | - |
| MiddlePhalanxOutlineCorrect | 2 | 1,700 | 2.65M | 59.8 | 67.4 | 12.7M | 58.1 | 69.8 | 9.56M | 61.2 | 67.7 | 0.3M | 91.1 |
| MiddlePhalanxTW | 6 | 1345 | 2.65M | 53.9 | 54.5 | 12.7M | 53.9 | 48.7 | 9.56M | 51.9 | 46.8 | 0.3M | 84.9 |
| OliveOil | 4 | 150 | 2.65M | 66.7 | 46.7 | 12.7M | 76.7 | 70.0 | 9.56M | 73.3 | 73.3 | 0.3M | - |
| PhalangesOutlinesCorrect | 2 | 2,785 | 2.65M | 62.2 | 65.4 | 12.7M | 63.9 | 67.5 | 9.56M | 62.7 | 67.5 | 0.3M | - |
| Plane | 7 | 1,424 | 2.65M | 33.3 | 81.0 | 12.7M | 39.0 | 78.1 | 9.56M | 38.1 | 83.8 | 0.3M | |
| PowerCons | 2 | 2,007 | 2.65M | 77.8 | 79.0 | 12.7M | 72.8 | 81.1 | 9.56M | 77.8 | 80.6 | 0.3M | - |
| ProximalPhalanxOutlineCorrect | 2 | 1,298 | 2.65M | 71.5 | 82.8 | 12.7M | 73.9 | 85.6 | 9.56M | 72.9 | 83.9 | 0.3M | - |
| ProximalPhalanxTW | 6 | 1,101 | 2.65M | 67.8 | 80.0 | 12.7M | 69.8 | 80.0 | 9.56M | 68.8 | 74.1 | 0.3M | - |
| SemgHandGenderCh2 | 4 | 2,840 | 2.65M | 49.1 | 54.7 | 12.7M | 59.5 | 67.2 | 9.56M | 58.3 | 73.3 | 0.3M | - |
| ShapeletSim | 2 | 1,353 | 2.65M | 71.3 | 69.5 | 12.7M | 76.1 | 59.4 | 9.56M | 82.2 | 54.4 | 0.3M | - |
| SmallKitchenAppliances | 2 | 2,207 | 2.65M | 66.2 | 69.3 | 12.7M | 60.8 | 63.2 | 9.56M | 57.6 | 61.6 | 0.3M | 83.5 |
| SonyAIBORobotSurface1 | 2 | 2,558 | 2.65M | 54.2 | 60.4 | 12.7M | 64.1 | 71.7 | 9.56M | 68.2 | 78.5 | 0.3M | - |
| SonyAIBORobotSurface2 | 2 | 2,596 | 2.65M | 54.8 | 69.9 | 12.7M | 55.9 | 70.6 | 9.56M | 55.1 | 63.8 | 0.3M | - |
| StarLightCurves | 3 | 27,131 | 2.65M | 67.8 | 72.9 | 12.7M | 68.6 | 72.6 | 9.56M | 67.6 | 70.1 | 0.3M | - |
| Strawberry | 2 | 3,593 | 2.65M | 71.2 | 85.1 | 12.7M | 69.5 | 84.9 | 9.56M | 69.5 | 88.4 | 0.3M | 97.6 |
| ToeSegmentation1 | 2 | 3,889 | 2.65M | 66.9 | 66.2 | 12.7M | 53.5 | 52.2 | 9.56M | 53.5 | 60.1 | 0.3M | - |
| ToeSegmentation2 | 2 | 2,714 | 2.65M | 79.7 | 73.1 | 12.7M | 69.2 | 59.2 | 9.56M | 77.7 | 80.0 | 0.3M | - |
| Trace | 4 | 870 | 2.65M | 49.5 | 88.0 | 12.7M | 54.0 | 90.0 | 9.56M | 47.0 | 77.0 | 0.3M | 100 |
| TwoLeadECG | 2 | 2,487 | 2.65M | 59.6 | 69.1 | 12.7M | 53.2 | 64.6 | 9.56M | 53.2 | 63.9 | 0.3M | 97.8 |
| UMD | 3 | 816 | 2.65M | 47.7 | 69.5 | 12.7M | 38.9 | 59.7 | 9.56M | 42.4 | 60.4 | 0.3M | - |
| Wafer | 2 | 4,805 | 2.65M | 94.6 | 96.8 | 12.7M | 91.3 | 93.5 | 9.56M | 90.9 | 95.2 | 0.3M | 100 |
| Wine | 2 | 171 | 2.65M | 53.6 | 57.4 | 12.7M | 59.3 | 63.0 | 9.56M | 63.0 | 55.6 | 0.3M | 90.7 |
| Worms | 5 | 5,377 | 2.65M | 62.6 | 67.5 | 12.7M | 57.1 | 64.9 | 9.56M | 54.5 | 63.6 | 0.3M | 83.1 |
| WormsTwoClass | 2 | 5377 | 2.65M | 74.3 | 81.8 | 12.7M | 62.3 | 70.1 | 9.56M | 61.0 | 79.2 | 0.3M | 98.7 |

In medical time series domains, ptb-db and MIT-BIH arrhythmia data sets are mostly used. EEG eye state data set has two categories,and because of its high complexity, the accuracy always stays

Table 10: Full comparison results of medical time series classification tasks(%) on EEG eye states, ptb-db and MIT-BIH.

| Data | Classes Number | Symbols Number | RoBERTa$_{Large}$ | | | Llama2-7B | | | Mistreal-7B | | | CNN | BiRNN | LSTM |
|---|---|---|---|---|---|---|---|---|---|---|---|---|---|---|
| | | | r=16 | r=64 | r=256 | r=16 | r=64 | r=256 | r=16 | r=64 | r=256 | | | |
| EEG | 2 | 938 | 60.1 | 66.0 | 64.4 | 55.9 | 57.4 | 57.5 | 58.5 | 58.0 | 60.1 | * | * | * |
| ptb-db | 2 | 2179 | 89.5 | 90.6 | 89.3 | 99.0 | 98.6 | 98.3 | 98.9 | 98.7 | 98.6 | 99.4 | * | * |
| mit-bih | 5 | 2926 | 86.4 | 86.4 | 86.3 | 89.6 | 89.4 | 89.1 | 89.3 | 89.7 | 89.3 | 93.4 | 96.5 | 88.1 |

Table 11: Full comparison results of the regression task on 19 Monashe Time Series Regression datasets.

| Data | Symbols Number | RoBERTa$_{Large}$ | | | Llama2-7B | | | Mistreal-7B | | | SOTA (cite) |
|---|---|---|---|---|---|---|---|---|---|---|---|
| | | r=16 RMSE | r=64 RMSE | r=256 RMSE | r=16 RMSE | r=64 RMSE | r=256 RMSE | r=16 RMSE | r=64 RMSE | r=256 RMSE | RMSE |
| AppliancesEnergy | 778 | 1.73 | 2.09 | 1.74 | 2.43 | 2.43 | 2.43 | 2.34 | 2.02 | 2.11 | 2.29 |
| HouseholdPowerConsumption1 | 1717 | 377.02 | 377.20 | 377.20 | 398.01 | 398.05 | 398.05 | 228.83 | 228.78 | 228.67 | 132.80 |
| HouseholdPowerConsumption2 | 1717 | 27.64 | 27.71 | 27.73 | 36.63 | 36.71 | 36.69 | 24.54 | 24.56 | 24.51 | 32.61 |
| BenzeneConcentration | 3037 | 4.01 | 4.00 | 4.00 | 5.57 | 5.56 | 5.56 | 4.03 | 4.03 | 4.03 | 0.64 |
| BeijingPM10Quality | 970 | 66.16 | 66.07 | 66.07 | 93.25 | 93.26 | 93.26 | 65.25 | 65.25 | 65.24 | 93.14 |
| BeijingPM25Quality | 970 | 54.16 | 54.16 | 54.16 | 76.75 | 76.73 | 76.73 | 53.50 | 53.49 | 53.49 | 59.50 |
| LiveFuelMoistureContent | 5689 | 20.56 | 20.56 | 20.56 | 29.32 | 29.33 | 29.32 | 20.94 | 20.88 | 20.85 | 29.41 |
| FloodModeling1 | 969 | 0.00 | 0.00 | 0.00 | 0.05 | 0.05 | 0.05 | 0.37 | 0.36 | 0.36 | 0.00 |
| FloodModeling2 | 979 | 0.00 | 0.00 | 0.00 | 0.05 | 0.04 | 0.04 | 0.40 | 0.39 | 0.39 | 0.01 |
| FloodModeling3 | 948 | 0.00 | 0.00 | 0.00 | 0.06 | 0.05 | 0.05 | 0.41 | 0.37 | 0.39 | 0.00 |
| AustraliaRainfall | 4740 | 4.36 | 4.36 | 4.36 | 6.05 | 6.01 | 6.02 | 4.31 | 4.28 | 4.30 | 8.12 |
| PPGDalia | 12298 | 9.32 | 9.32 | 9.32 | 12.54 | 12.50 | 12.52 | 9.04 | 9.02 | 9.03 | 9.92 |
| IEEEPPG | 8971 | 17.06 | 17.00 | 17.04 | 22.59 | 22.53 | 22.55 | 17.15 | 17.12 | 17.16 | 23.90 |
| BIDMC32HR | 9423 | 6.73 | 6.98 | 6.71 | 12.02 | 11.98 | 12.04 | 8.24 | 8.21 | 8.23 | 9.42 |
| BIDMC32RR | 9412 | 1.77 | 1.74 | 1.76 | 2.64 | 2.61 | 2.62 | 2.09 | 2.06 | 2.08 | 3.02 |
| BIDMC32SpO2 | 5537 | 2.90 | 2.85 | 2.89 | 3.82 | 3.79 | 3.81 | 2.95 | 2.91 | 2.93 | 4.45 |
| NewsHeadlineSentiment | 5537 | 0.07 | 0.07 | 0.07 | 0.13 | 0.13 | 0.13 | 0.11 | 0.11 | 0.11 | 0.14 |
| NewsTitleSentiment | 5537 | 0.07 | 0.07 | 0.07 | 0.13 | 0.13 | 0.13 | 0.11 | 0.11 | 0.11 | 0.14 |
| Covid3Month | 227 | 0.02 | 0.02 | 0.02 | 0.11 | 0.11 | 0.11 | 0.45 | 0.44 | 0.44 | 0.04 |

at around 60%. EEG eye state data and MIT-BIH has more than one channel, which indicates that LLM-ABBA might have the ability to process complicate features across channels. Table 10 presents the full medical time series classification results using LLM-ABBA.

LLM-ABBA achieves comparable time series prediction results to SOTAs, and there is no over-fitting in these tasks when using different low rank $r$. Because that ABBA tends to symbolize trends and altitudes of the time series signals, LLM-ABBA always strengthen the vibration of predicted time series segments which can be seen in Figure 7.

Table 12: Full comparison results of the prediction task on 4 time series prediction datasets.

| Data | Predictor Length | Symbols Number | Llama2-7B | | | | | | Mistreal-7B | | | | | |
|---|---|---|---|---|---|---|---|---|---|---|---|---|---|---|
| | | | r=16 | | r=64 | | r=256 | | r=16 | | r=64 | | r=256 | |
| | | | MSE | MAE | MSE | MAE | MSE | MAE | MSE | MAE | MSE | MAE | MSE | MAE |
| ETTh1 | 168/24 | 2789 | 0.689 | 0.653 | 0.647 | 0.696 | 0.658 | 0.677 | 0.631 | 0.681 | 0.622 | 0.631 | 0.626 | 0.677 |
| ETTh2 | 168/24 | 5383 | 0.798 | 0.788 | 0.784 | 0.761 | 0.789 | 0.772 | 0.776 | 0.787 | 0.759 | 0.761 | 0.762 | 0.771 |
| ETTm1 | 168/24 | 3170 | 0.403 | 0.397 | 0.386 | 0.364 | 0.392 | 0.385 | 0.457 | 0.422 | 0.401 | 0.387 | 0.407 | 0.397 |
| ETTm2 | 168/24 | 6878 | 0.224 | 0.209 | 0.201 | 0.198 | 0.215 | 0.207 | 0.251 | 0.237 | 0.214 | 0.203 | 0.218 | 0.209 |
| ETTh1 | 168/96 | 2789 | 0.762 | 0.786 | 0.754 | 0.752 | 0.759 | 0. 60 | 0.792 | 0.804 | 0.773 | 0.782 | 0.781 | 0.788 |
| ETTh2 | 168/96 | 5383 | 0.912 | 0.885 | 0.892 | 0.881 | 0.907 | 0.876 | 0.899 | 0.887 | 0.871 | 0.866 | 0.878 | 0.872 |
| ETTm1 | 168/96 | 3170 | 0.542 | 0.537 | 0.531 | 0.528 | 0.538 | 0.520 | 0.541 | 0.533 | 0.524 | 0.517 | 0.529 | 0.520 |
| ETTm2 | 168/96 | 6878 | 0.302 | 0.286 | 0.288 | 0.267 | 0.293 | 0.278 | 0.289 | 0.302 | 0.276 | 0.281 | 0.280 | 0.285 |
| ETTh1 | 168/168 | 2789 | 1.161 | 1.010 | 1.087 | 0.964 | 1.096 | 0.989 | 1.182 | 1.217 | 1.174 | 1.968 | 1.179 | 1.992 |
| ETTh2 | 168/168 | 5383 | 4.103 | 2.675 | 3.975 | 2.101 | 4.086 | 2.537 | 4.092 | 2.626 | 3.898 | 2.134 | 3.910 | 2.245 |
| ETTm1 | 168/168 | 3170 | 0.989 | 0.962 | 0.974 | 0.952 | 0.979 | 0.959 | 1.001 | 0.986 | 0.966 | 0.958 | 0.972 | 0.966 |
| ETTm2 | 168/168 | 6878 | 0.616 | 0.583 | 0.576 | 0.544 | 0.580 | 0.561 | 0.592 | 0.541 | 0.521 | 0.503 | 0.532 | 0.509 |

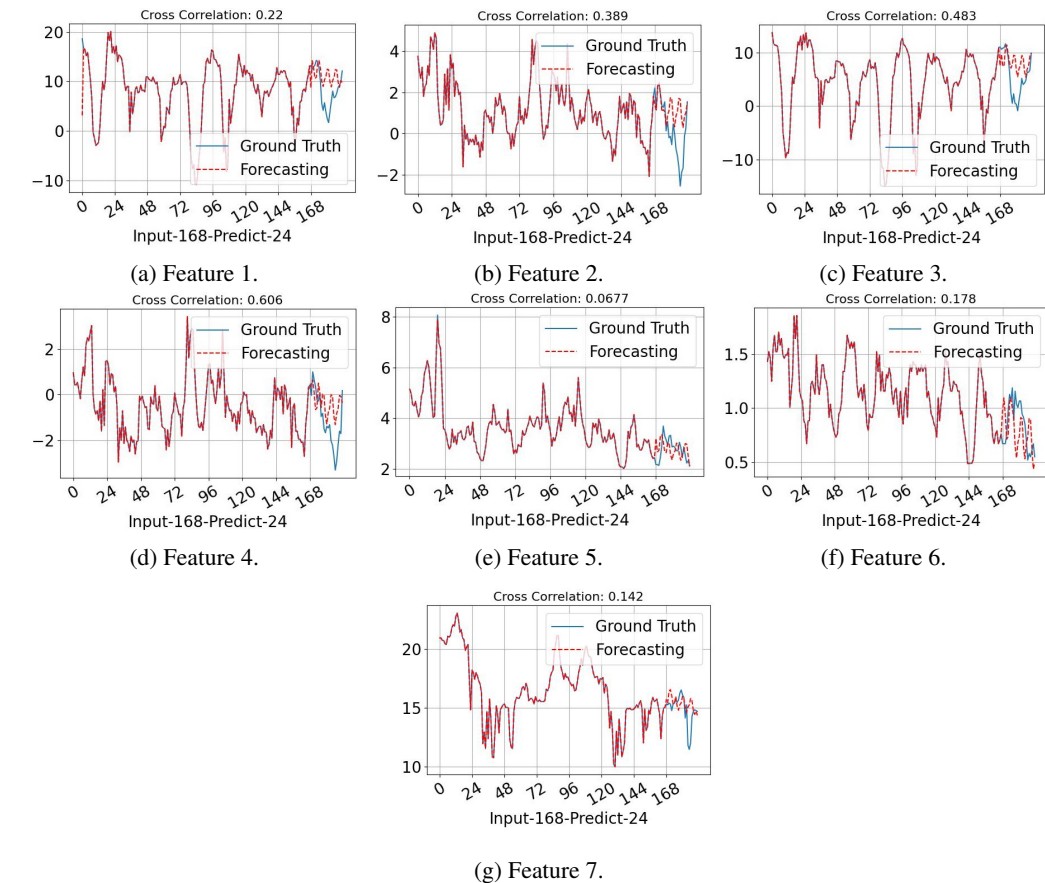

Figure 7: Visualization of input-168-predict-24 results on ETTh1 by using LLM-ABBA .

Table 13: The performance of LLM-ABBA with extra new tokens on ETTh1 data in terms of time series forecasting tasks.

| Data | Predictor Length | Symbols Number | MSE | MAE | MSE | MAE | MSE | MAE |
|------|---------|---------|------|------|------|------|------|------|
| | | | r=16 | | r=64 | | r=256 | |
| ETTh1 | 168/24 | 2789 | 0.636 | 0.692 | 0.626 | 0.632 | 0.629 | 0.681 |
| ETTh2 | 168/24 | 5383 | 0.779 | 0.788 | 0.761 | 0.763 | 0.763 | 0.777 |
| ETTm1 | 168/24 | 3170 | 0.457 | 0.402 | 0.402 | 0.387 | 0.407 | 0.399 |
| ETTm2 | 168/24 | 6878 | 0.253 | 0.238 | 0.215 | 0.203 | 0.219 | 0.209 |

## G  SEMANTIC CONSISTENCY

More or less, when using pretrained tokens as the input symbols, fine-tuning on no language content (such as time series signals) will bring semantic loss to LLMs. Therefore, we use ASCII codes to generate new symbols by adding more digits and expand the used alphabet table. Following the same fine-tuning process to the above experiment settings, we compute the forecasting performance by fine-tuing on Mistral-7B. Compared to Table 12, Table 13 shows that the difference is not noticeable. We also evaluate the fine-tuned Mistral-7B on five mostly asked questions of ChatGPT in terms of five different domains. Eventually, we find that LLM-ABBA has no semantic loss, which can be seen in Table 14, 15, 16, 17, 18.

## H  LIMITATIONS

Our proposed FAPCA strategy to ABBA still cannot fully guarantee a completely cumulative error arisen from the previous mistaken symbols from the recovery because the mistaken $len_i$ often leads

Table 14: In weather prediction domain, users often ask for general information about the weather. We evaluate **Questions: "How do hurricanes form?"** on Mistral-7B. Semantic outputs between Mistral-7B and fine-tuned Mistral-7B are almost the same.

| Models | Answes |
|---|---|
| Mistral-7B | Hurricanes, also known as typhoons or cyclones in different parts of the world, are powerful storm systems that form over tropical oceans. They typically develop from tropical disturbances, which are clusters of thunderstorms that form over warm water. |
| | There are several conditions that must be met for a hurricane to form: |
| | 1. Warm water: Hurricanes usually form over waters that are at least 80 degrees Fahrenheit (26.5 degrees Celsius). This is considered to be the "fuel" that powers the storm. |
| | 2. Low atmospheric pressure: Hurricanes require areas of low pressure in the atmosphere, where air rises and forms clouds and thunderstorms. |
| | 3. Moisture: Humid air is also necessary for the formation of a hurricane. As the warm air rises, it cools and condenses, forming clouds and releasing heat that powers the storm. |
| | 4. Wind patterns: The wind patterns in the atmosphere must also be conducive to the formation of a hurricane. The winds must converge near the surface of the ocean, helping to form the area of low pressure, while upper-level winds must be favorable and not too strong to allow the storm to develop and grow. |
| | When all these conditions are met, a cluster of thunderstorms can form over the ocean, and as the warm air rises and more air is drawn into the system, the storm can strengthen and grow, eventually leading to the formation of a hurricane. The hurricane will continue to intensify |
| Mistral-7B fine-tuned on ETTh1 | Hurricanes, also known as typhoons or cyclones in different parts of the world, are powerful storm systems that form over tropical oceans. They typically develop from tropical disturbances, which are clusters of thunderstorms that form over warm water. |
| | There are several conditions that must be met for a hurricane to form: |
| | 1. Warm water: Hurricanes usually form over waters that are at least 80 degrees Fahrenheit (26.5 degrees Celsius). This is considered to be the "fuel" that powers the storm. |
| | 2. Low atmospheric pressure: Hurricanes require areas of low pressure in the atmosphere, where air rises and forms clouds and thunderstorms. |
| | 3. Moisture: Humid air is also necessary for the formation of a hurricane. As the warm air rises, it cools and condenses, forming clouds and releasing heat that powers the storm. |
| | 4. Wind patterns: The wind patterns in the atmosphere must also be conducive to the formation of a hurricane. The winds must converge near the surface of the ocean, helping to form the area of low pressure, while upper-level winds must be favorable and not too strong to allow the storm to develop and grow. |
| | When all these conditions are met, a cluster of thunderstorms can form over the ocean, and as the warm air rises and more air is drawn into the system, the storm can strengthen and grow, eventually leading to the formation of a hurricane. The hurricane will continue to intensify |

to a minor shift if certain symbols are replaced improperly. Additionally, as a congenital defect of LLMs, hallucination cannot be addressed in this work, and the vibration or adverse response of predicted sequence cannot be completely solved. Moreover, after using ABBA to transform time series, most LLMs only can support up to 4,096 tokens, which fundamentally abandons the long-term time series analysis tasks.

Table 15: For the fact checking questions, we ask Mistral-7B **Question: "How long does it take to travel to Mars?"** Mistral-7B can provide accurate and brief information from a variety of sources.

| Models | Answers |
|---|---|
| Mistral-7B | It takes about 6-9 months for a spacecraft traveling at the speed of light (186,250 miles per second) from Earth to reach its closest point in our solar system. |
| Mistral-7B fine-tuned on ETTh1 | It takes about 6-9 months for a spacecraft traveling at the speed of light (186,250 miles per second) from Earth to reach its closest point in our solar system. |

Table 16: In the technology explanation domain, **Question: "What is quantum computing?"** is asked.

| Models | Answers |
|---|---|
| Mistral-7B | Quantum computing is a type of computing that uses quantum-mechanical phenomena, such as superposition and entanglement, to perform operations on data. Unlike classical computers, which store and process information using bits (0s and 1s), quantum computers use quantum bits, or qubits, which can exist in multiple states simultaneously. This allows quantum computers to perform certain types of calculations much faster than classical computers. |
| Mistral-7B fine-tuned on ETTh1 | Quantum computing is a type of computing that uses quantum-mechanical phenomena, such as superposition and entanglement, to perform operations on data. Unlike classical computers, which store and process information using bits (0s and 1s), quantum computers use quantum bits, or qubits, which can exist in multiple states simultaneously. This allows quantum computers to perform certain types of calculations much faster than classical computers. |

Table 17: In philosophical domains, **Question: "What is consciousness?"** is evaluated. Although these questions often do not have clear answers, Mistral-7B can output the same philosophical approaches and ideas.

| Models | Answers |
|---|---|
| Mistral-7B | Consciousness is the state of being aware of one's surroundings and experiences, including thoughts, emotions, and perceptions. It is a subjective experience that cannot be directly observed or measured, but can be inferred from behavior and brain activity. The nature of consciousness remains a topic of debate and research in philosophy, psychology, and neuroscience. |
| Mistral-7B fine-tuned on ETTh1 | Consciousness is the state of being aware of one's surroundings and experiences, including thoughts, emotions, and perceptions. It is a subjective experience that cannot be directly observed or measured, but can be inferred from behavior and brain activity. The nature of consciousness remains a topic of debate and research in philosophy, psychology, and neuroscience. |

Table 18: In digital health domains, **Question: "What are the symptoms of the flu?"** is asked.

| Models | Answers |
|---|---|
| Mistral-7B | The flu is a viral infection that can cause a wide range of symptoms. Some common symptoms include fever, chills, cough, sore throat, runny or stuffy nose, body or muscle aches, headaches, and fatigue. Symptoms typically appear one to three days after exposure to the virus and can last for seven to ten days. However, some people may have a milder illness or no symptoms at all. |
| Mistral-7B fine-tuned on ETTh1 | The flu is a viral infection that can cause a wide range of symptoms. Some common symptoms include fever, chills, cough, sore throat, runny or stuffy nose, body or muscle aches, headaches, and fatigue. Symptoms typically appear one to three days after exposure to the virus and can last for seven to ten days. However, some people may have a milder illness or no symptoms at all. |

