# OpenReview forum: "LLM-ABBA: Fine-Tuning Large Language Models For Time Series Using Symbolic Approximation"
_ICLR.cc/2025/Conference — ICLR 2025 Conference Withdrawn Submission_

### Official Review · Reviewer_5LtN · 2024-10-28

**Soundness:** 2
**Presentation:** 2
**Contribution:** 1
**Rating:** 3
**Confidence:** 5

**Summary:**

The authors propose LLM-ABBA, a method that integrates symbolic time-series approximation with large language models (LLMs). The proposed method first encodes time-series features into tuples that contain the lengths and increments of short time-series sequences. Then, the tuples are discretized using clustering methods. The cluster centers are transformed into characters (in text) to produce a symbolic representation of the time-series data. For specific time-series analysis tasks and datasets, the authors fine-tune the LLMs using QLoRA to make predictions. Experiments are conducted with three LLMs (RoBERT, Llama2-7B, and Mistral-7B) on three time-series tasks: classification, regression, and forecasting.

**Strengths:**

1. The technical content is easy to follow.
2. This paper proposes a method to connect conventional symbolic aggregation methods in time-series with LLMs, which could potentially be helpful in tokenizing time-series data to be processed by LLMs.
3. Experiments are conducted with three LLMs, which help validate the effectiveness of the proposed framework.

**Weaknesses:**

This paper has critical weaknesses from several perspectives:

> Benchmarking

- The baselines are not comprehensive. The proposed method uses LLMs as one of the key components, but the baselines are mostly non-LLM methods. There are several notable methods that also use pre-trained LLMs that should be compared. For example, [1] uses reprogramming (only mentioned in forecasting but ignored on other tasks), [2] fine-tunes some layers in the LLMs, and [3] performs forecasting in a zero-shot manner by transforming real-valued time series into text, which is very related to the proposed method.
- The benchmarking for classification is not appropriate. The authors only compared their method with the "SOTA" method TS2Vec. Although TS2Vec demonstrates top-tier performance, comparing with only one method does not provide sufficient information to assess the performance of the proposed method. Comparisons with more baselines are necessary to provide an unbiased view, as baseline methods may perform differently on different datasets (e.g., TS2Vec is outperformed by other baselines on many datasets).
- The benchmarking results are incomplete and inconsistent. In Table 1 and Table 9, there are many missing values in the SOTA column. However, most of these values are reported in the TS2Vec paper. Additionally, some provided results are inconsistent with the TS2Vec paper. The benchmarking setups (e.g., how the results were obtained—were they reproduced?) should be included but are missing. Furthermore, Table 2 also covers classification datasets but uses a different set of baselines, while the "SOTA" method TS2Vec is ignored. The inconsistency between Table 1 and Table 2 is very confusing.
- Table 3 reports results on regression tasks, but the "SOTA" method for these results is not specified. From [4], it seems that the "SOTA" column includes the best results from several conventional methods, including Random Forest, XGBoost, Rocket, FCN, ResNet, etc. Although this provides better baselines compared to the classification tasks, more recent and advanced methods should be considered. According to [4], time-series regression is closely related to classification, where SOTA classification methods (such as TS2Vec, TimesNet, T-Rep, DTW, etc.) should be evaluated on the regression datasets and compared.
- Table 4 includes more baselines but has many missing values, and the reasons for these missing values are not discussed. Many of the methods are open-sourced. Additionally, Informer is not a SOTA method for forecasting. More recent baselines (including TimesNet, iTransformer, TimeMixer, DLinear, PatchTST, MOMENT, etc.) should be compared. The implementations are publicly available (see [this repo](https://github.com/thuml/Time-Series-Library)) so they should be easy to reproduce.

> Ablation study is missing

No ablation study is conducted in this paper, which leads to two essential concerns:
1. The benefit of using LLMs cannot be justified. The proposed method uses QLoRA for adapting LLMs to time-series tasks. However, since QLoRA still involves training a significant number of parameters, it is unclear whether the predictive power comes from the LLMs or the QLoRA component. Therefore, an ablation case where predictions are based only on the ABBA features (i.e., without using LLMs) should be included.
2. [3] also proposes a method of translating time series into text that can be processed by LLMs in a zero-shot manner. The major difference in this paper is that the authors use ABBA to perform this translation. The effectiveness of using ABBA compared to the direct translation in [3] cannot be validated without an ablation study that starts with zero-shot or few-shot settings, instead of directly using the expensive QLoRA.

> Novelty and soundness

1. The novelty of the proposed method is limited. Section 3.1 is directly obtained from ABBA (Elsworth & Gütell, 2020). Section 3.2 discusses the procedure of feeding the ABBA features to LLMs with task information and instructions, which is not novel, as a similar approach is presented in [1]. Section 3.3 presents a simple approach that modifies the relative increment in ABBA to absolute magnitude, which seems to improve the recovery error based on Figure 4.
2. The soundness is questionable. From Section 3.1.2 and Figure 3, the ABBA cluster centers are translated into characters, which can include special characters. However, LLMs trained on natural language may not have the ability to produce meaningful outputs given a sequence of these characters. Therefore, the purpose of using LLMs in this context is not sound, which should be better validated through ablation study.

In conclusion, the main claims are not supported by the experiments or are not justified, including:
1. Contribution 1: "Enabling an effective inference task over out-of-sample data" is not demonstrated in the experiments.
2. Contribution 2: Due to the benchmarking issues mentioned above, the performance compared to SOTA methods cannot be concluded.
3. Contribution 4: This is not supported or justified by the experiments.

### Reference
[1] Jin et al., Time-LLM: Time Series Forecasting by Reprogramming Large Language Models, ICLR 2024.

[2] Zhou et al., One Fits All: Power General Time Series Analysis by Pretrained LM, NeurIPS 2023.

[3] Gruver et al., Large Language Models are Zero-Shot Time Series Forecasters, NeurIPS 2023.

[4] Tan et al., Time Series Extrinsic Regression, KDD 2021.

**Questions:**

1. Why are there many missing values in Table 1, Table 2, and Table 4?
2. In multivariate time series, are the symbols defined for each univariate time series separately, or can different variates share the same symbol?
3. Some datasets require a large volume of symbols. On average, how many symbols are needed to describe a time series sample?

---

### Official Review · Reviewer_nyR4 · 2024-11-03

**Soundness:** 2
**Presentation:** 2
**Contribution:** 2
**Rating:** 3
**Confidence:** 3

**Summary:**

This article proposes to use pre-trained large language models to perform various tasks (classification, prediction, extrinsic regression) with time series. The authors use the ABBA method to approximate time series and convert them into symbolic sequences and QLoRA to fine-tune models.

**Strengths:**

LLM for time series is a hot topic with many possible applications. Since time series and text data are widely different modalities, methodologies that can bridge the gap between the two are interesting.

**Weaknesses:**

- As I understand it, the contribution is limited: it mainly combines ABBA, a symbolization method, and QLoRA to fine-tune pre-trained LLMs.
- The empirical results do not show a consistent improvement in most tasks. A systematic analysis of why it works will help readers understand if LLM-ABBA is helpful for their use case.
- It would be more convincing to present experiment results with the standard deviation over several evaluations (all presented algorithms have randomness because of the stochastic optimization, the train/test splits, etc.)
- The presentation could be improved: a large part of the article describes background methods, and the actual methodological contribution is only described in sections 3.2 and 3.3 (one page in total). However, several important pieces of information (especially in the experiments) are deferred to the appendices because of the lack of space.

**Questions:**

- Can the authors identify significant trends from the empirical results? Is LLM-ABBA better on long-time series or short-time series? Is it robust to noise or time-warping? Why does LLM-ABBA seem better suited for the regression task?
- Unless I am mistaken, it is not specified where the SOTA scores come from. Can the authors be specific on that part?
- How was ABBA calibrated?
- Related question: How does ABBA’s calibration influence the downstream tasks? (In other words, what happens when the parameters of ABBA change?)

---

### Official Review · Reviewer_xswM · 2024-11-04

**Soundness:** 3
**Presentation:** 3
**Contribution:** 1
**Rating:** 3
**Confidence:** 4

**Summary:**

The authors propose LLM-ABBA, a method that approximates time series with a symbolic representation which is subsequently used to fine tune LLMs on time series task such as regression, classification, and forecasting. For the approximation they rely on the adaptive Brownian bridge-based symbolic aggregation (ABBA) method and slightly extend it to mitigate the negative effect a wrongly chosen token can have on the symbol-base prediction. On various time series data sets, the authors show mixed results of their model's performance when comparing it to SOTA models.

**Strengths:**

The authors evaluate their method on a large variety of data sets and time series tasks. They also spend a significant amount of the manuscript to describe the inner workings of the ABBA algorithm. Overall, the paper is comprehensible and relatively easy to follow.

**Weaknesses:**

I miss the comparison with other LLM-based time series models that are mentioned in the introduction. While the comparison with SOTA on any given task is warranted, it'd be important to compare their method to other LLM-based methods. Additionally, the paper lacks a comparison/discussion with/of other symbolic approximations such as SAX[1]. What is the motivation for using ABBA over SAX or other approximations? A table which compares different approximations used in differently sized models could help the community to better understand the value of such symbolic approximations w.r.t LLM architectures. I also miss statistical rigor in the comparison tables. In particular, computing standard deviations (e.g., via bootstrapping) to obtain the statistical significance of the results is important. Showing a critical difference plot[2] could be a good first step. Overall, the manuscript is a good starting point for deeper investigations of the space of symbolic approximations of TS and LLMs. However, at the current level of depth and statistical rigor, it does not meet the bar of this venue.


1. Lin, J., Keogh, E., Wei, L., & Lonardi, S. (2007). Experiencing SAX: a novel symbolic representation of time series. Data Mining and knowledge discovery, 15, 107-144.
2. Demšar, J. (2006). Statistical comparisons of classifiers over multiple data sets. The Journal of Machine learning research, 7, 1-30.

**Questions:**

1. Why did you choose ABBA over other approximations?
2. Can you compute the statistical significance of your results (or show a critical difference plot)?
3. Did you observe time series characteristics with which ABBA-LLM struggles/performs well?
4. Figure 3: What is the final approach that you choose? I assume the right panel. If this is the case, I'd add a more verbose caption to the Figure to clarify this.
5. Can you motive in more detail (formalized) the problem you describe in 3.3 and how your approach solves it? In my opinion this is your only technical contribution which should come out stronger than it currently is.
6. Table 1: It seems like some data sets do not have a SOTA value, why is that?
7. In Section 4.3, I don't understand the last sentence of the second paragraph. Could you clarifiy?
8. How sensitive is the approach to the parameters and how did you arrive at the values that you report? Especially, given that some of the data sets you evaluate on don't have an evaluation split.

---

### Note · Authors · 2024-11-26

I have read and agree with the venue's withdrawal policy on behalf of myself and my co-authors.